# Differential regulation of cranial and cardiac neural crest by serum response factor and its cofactors

**Colin J Dinsmore, Philippe Soriano\***

Department of Cell, Development and Regenerative Biology, Icahn School of Medicine at Mount Sinai, New York, United States

**Abstract** Serum response factor (SRF) is an essential transcription factor that influences many cellular processes including cell proliferation, migration, and differentiation. SRF directly regulates and is required for immediate early gene (IEG) and actin cytoskeleton-related gene expression. SRF coordinates these competing transcription programs through discrete sets of cofactors, the ternary complex factors (TCFs) and myocardin-related transcription factors (MRTFs). The relative contribution of these two programs to in vivo SRF activity and mutant phenotypes is not fully understood. To study how SRF utilizes its cofactors during development, we generated a knock-in $Srf^{al}$ allele in mice harboring point mutations that disrupt SRF-MRTF-DNA complex formation but leave SRF-TCF activity unaffected. Homozygous $Srf^{al/al}$ mutants die at E10.5 with notable cardiovascular phenotypes, and neural crest conditional mutants succumb at birth to defects of the cardiac outflow tract but display none of the craniofacial phenotypes associated with complete loss of SRF in that lineage. Our studies further support an important role for MRTF mediating SRF function in cardiac neural crest and suggest new mechanisms by which SRF regulates transcription during development.

## Editor's evaluation

This carefully executed study demonstrates new mechanisms by which Serum Response Factor (Srf) regulates transcription. The authors report the effects that loss of Srf function has on different neural crest lineages in the mouse. The results convincingly show that the main function of Srf within neural crest is in the cardiac neural crest lineage where it regulates cytoskeletal genes.

**\*For correspondence:**
philippe.soriano@mssm.edu

**Competing interest:** The authors declare that no competing interests exist.

## Introduction

Multicellular development requires the precise management of cellular behaviors including proliferation, migration, and differentiation. These are coordinated through intercellular communication pathways, such as growth factor signaling, that couple extracellular information with internal effectors, including transcription factors (TFs) (*Fantauzzo and Soriano, 2015*; *Lemmon and Schlessinger, 2010*). The balance between opposing transcription programs is tuned by signaling pathways which activate specific TFs or in some cases cofactors that direct the behavior of a common TF. One example of the latter is the essential transcription factor Serum Response Factor (SRF) (*Posern and Treisman, 2006*). SRF is necessary for the expression of immediate early genes (IEGs) in cells stimulated with serum or growth factors, as well as many genes related to the actin cytoskeleton, contractility, and muscle differentiation.

SRF binds a conserved DNA regulatory sequence known as a CArG box, a motif found at many cytoskeletal and growth-factor inducible gene promoters (*Mohun et al., 1991*; *Norman et al., 1988*; *Sun et al., 2006b*). SRF can, however, effect at least two unique transcriptional programs by coupling

with two families of cofactors that compete for a common binding site on SRF itself (*Miano, 2003*; *Wang et al., 2004 Figure 1A*). The ternary complex factors (TCFs) are E26 transformation-specific (ETS) family proteins activated by extracellular signal-regulated kinase 1/2 (ERK1/2) phosphorylation (*Mylona et al., 2016*). Once activated, they bind DNA and promote cellular proliferation by transcribing IEGs in coordination with SRF (*Esnault et al., 2017*; *Gualdrini et al., 2016*). There are three TCF members in mouse and human: ELK1, ELK3/NET, and ELK4/SAP1 (*Posern and Treisman, 2006*). Opposing SRF-TCF activity are the Myocardin Related Transcription Factors (MRTFs). These cofactors rely on SRF to bind DNA, promote cytoskeletal gene expression, and are particularly important in muscle differentiation (*Posern and Treisman, 2006*). MRTFs bind to and are inhibited by G-actin. Polymerization of G-actin into F-actin liberates MRTFs to translocate to the nucleus and bind SRF (*Miralles et al., 2003*). This can be promoted by multiple signaling pathways, including phosphoinositide 3-kinase (PI3K), that stimulate guanine nucleotide exchange factors to activate F-actin-promoting Rho-family GTPases (*Brachmann et al., 2005*; *Hanna and El-Sibai, 2013*; *Jiménez et al., 2000*; *Vasudevan and Soriano, 2014*). MRTFs are also positively and negatively regulated by extensive phosphorylation (*Panayiotou et al., 2016*). Three MRTFs are known to interact with SRF: Myocardin itself, MRTF-A/MKL1/MAL, and MRTF-B (*Parmacek, 2007*). *Myocd* is expressed specifically in muscle while *Mrtfa* and *Mrtfb* are more broadly expressed (*Posern and Treisman, 2006*). A fourth MRTF, MAMSTR/MASTR, interacts with MEF2 proteins and is not known to bind SRF (*Creemers et al., 2006*).

Srf and its cofactors have been extensively studied genetically. $Srf^{-/-}$ mutant mice die between E6.5 and E8.5 showing defects in mesoderm formation (*Arsenian et al., 1998*; *Niu et al., 2005*). Cofactor knockouts are comparatively mild. Single TCF mutants are all fully or partially viable (*Ayadi et al., 2001*; *Cesari et al., 2004*; *Costello et al., 2004*; *Weinl et al., 2014*) and *Elk1; Elk3; Elk4* triple null embryos have not been described in detail but survive until E14.5 without obvious defects (*Costello et al., 2010*; *Gualdrini et al., 2016*). $Mrtfa^{-/-}$ mutant mice are viable (*Li et al., 2006*; *Sun et al., 2006a*) whereas $Mrtfb^{-/-}$ mice are inviable between E13.5-E15.5, exhibiting cardiovascular defects (*Li et al., 2012*; *Oh et al., 2005*). $Myocd^{-/-}$ mice have the most severe phenotype and die at E10.5, also from cardiovascular defects (*Espinoza-Lewis and Wang, 2014*; *Li et al., 2003*). *Mrtfa; Mrtfb* double null mice have not been described, but conditional double mutants have shown these factors exhibit redundancy and broadly phenocopy loss of *Srf* in several tissues and cell types (*Cenik et al., 2016*; *Guo et al., 2018*; *Li et al., 2005a*; *Trembley et al., 2015*). However, studies comparing *Srf* and *Mrtfa; Mrtfb* mutants are not always identical. In megakaryocytes, loss of *Mrtfa* and *Mrtfb* is more severe than loss of *Srf* and there are large gene expression differences in the two models (*Smith et al., 2012*; *Halene et al., 2010*). Indeed, there is evidence that MRTFs may regulate genes independent of SRF or act as cofactors for TFs other than SRF (*Asparuhova et al., 2011*; *Kim et al., 2017*). Whether the differences in *Srf* versus *Mrtfa; Mrtfb* loss-of-function studies are due to SRF-TCF activity, SRF-independent MRTF activity, or TCF/MRTF-independent SRF activity remains uncertain. These studies are summarized in *Supplementary file 1*.

One tissue in which SRF was found to be essential is the neural crest (NC) (*Newbern et al., 2008*; *Vasudevan and Soriano, 2014*). The NC is a transient developmental population of cells that arises from the dorsal neural tube, migrates ventrally throughout the embryo, and gives rise to numerous cell types including the bone and connective tissue of the face, as well as smooth muscle cells in the cardiac outflow tract (*Bronner and Simões-Costa, 2016*). The extensive migration, proliferation, and various differentiation outcomes these cells undergo requires accurate coordination, and decades of study have revealed a panoply of signaling pathways and transcription factors important in these processes, including the fibroblast growth factor (FGF) pathway, platelet-derived growth factor (PDGF) pathway, and SRF itself (*Brewer et al., 2015*; *Dinsmore and Soriano, 2018*; *Newbern et al., 2008*; *Rogers and Nie, 2018*; *Tallquist and Soriano, 2003*; *Vasudevan and Soriano, 2014*). We and others have previously shown that *Srf* is required in the NC for craniofacial and cardiovascular development (*Newbern et al., 2008*; *Vasudevan and Soriano, 2014*). Intriguingly, mice homozygous for a hypomorphic allele of *Mrtfb* die shortly after birth with cardiac outflow tract defects and can be rescued by a neural crest-specific transgene (*Li et al., 2005b*). Assays in mouse embryonic palatal mesenchyme cells (MEPMs) indicated that stimulation with the secreted ligands fibroblast growth factor (FGF) or platelet-derived growth factor (PDGF) promoted SRF-TCF complex formation, but only PDGF promoted SRF-MRTF interactions, in a PI3K-dependent manner (*Vasudevan and Soriano, 2014*). Supporting the importance of SRF-MRTF interactions, *Srf* interacted genetically with *Pdgfra*

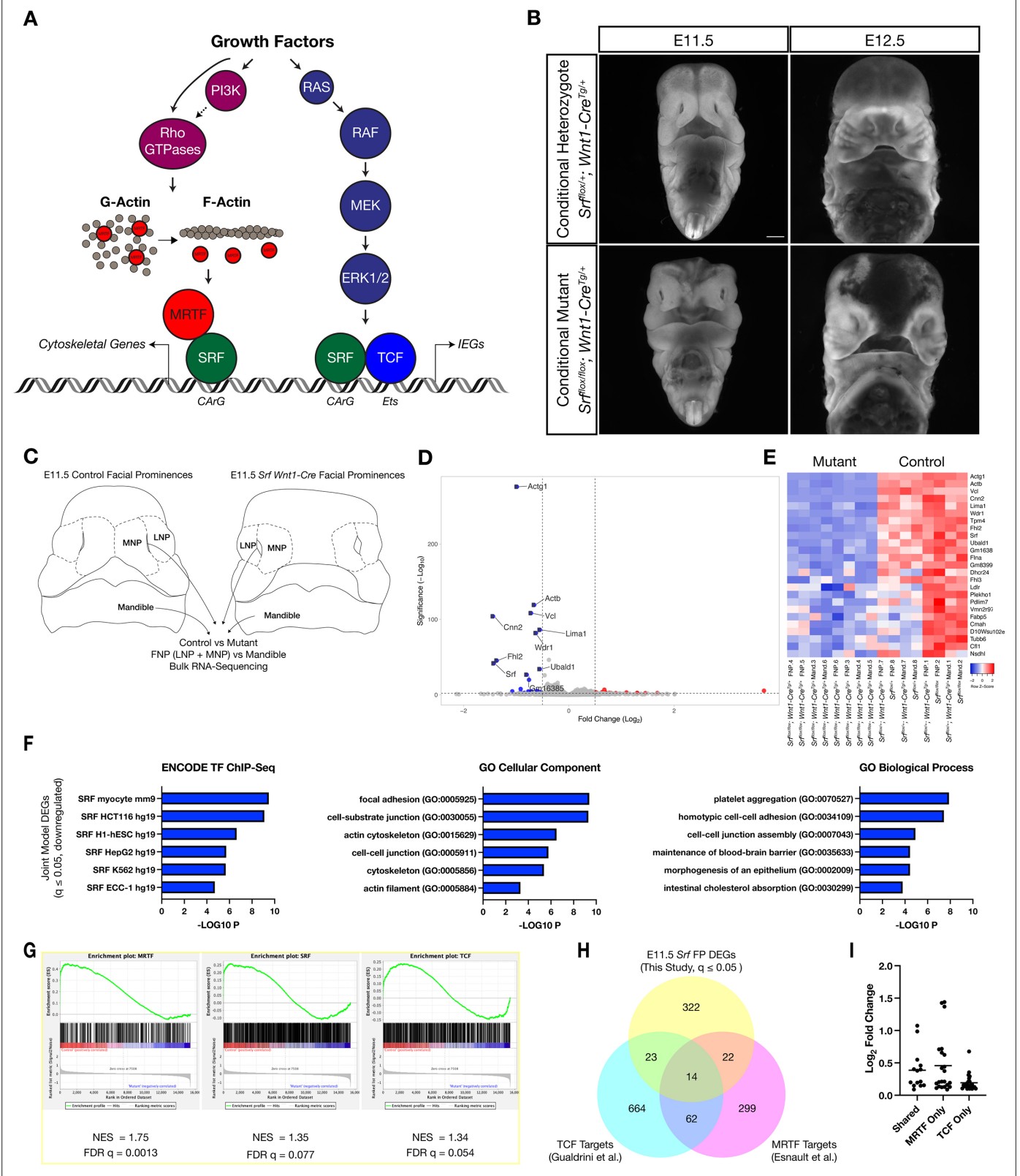

**Figure 1.** Loss of Srf in NC affects cytoskeletal gene expression. (**A**) Diagram depicting SRF, its TCF and MRTF cofactors, and the upstream signals that regulate them. (**B**) DAPI stained embryos at E11.5 and E12.5 show a facial cleft following loss of *Srf* in NC. Scale bar represents 1 mm. (**C**) Diagram depicting RNA-sequencing strategy. (**D**) Volcano plot showing DEGs in *Srf* NC conditional mutants. Genes with a *P*-value < 0.01 and log₂ fold change (FC) >0.25 are colored. Select genes are labeled. (**E**) A heatmap of the top 25 DEGs by q value. The samples cluster by genotype and are color-coded

*Figure 1 continued on next page*

*Figure 1 continued*

by Z-score. (**F**) Gene set enrichment analysis (GSEA) using a list of DEGs with q ≤ 0.05 and Log2FC ≤−0.25. Enrichment for ENCODE TF ChIP-Seq, GO Cellular Component, and GO Biological Process are shown. (**G**) GSEA for known SRF, MRTF, and TCF ChIP targets from previous datasets (*Esnault et al., 2014*; *Gualdrini et al., 2016*) across our entire dataset. (**H**) Overlap of known MRTF and TCF targets with DEGs q ≤ 0.05. (**I**) Absolute value of $log_2$ FC for DEGs that overlap with each category. Horizontal bar indicates the mean (0.385 Shared, 0.378 MRTF, 0.219 TCF).

The online version of this article includes the following figure supplement(s) for figure 1:

**Figure supplement 1.** Srf *NC* conditional mutants do not show early patterning defects.

**Figure supplement 2.** Additional data related to craniofacial *RNA-Seq*.

but not *Fgfr1* in NC (*Vasudevan and Soriano, 2014*). However, the contributions of each SRF-cofactor transcriptional program to the overall *Srf* NC phenotype are unclear.

In this study, we further characterize the molecular consequences of losing *Srf* expression in the NC through marker analysis and expression profiling, finding the most highly downregulated genes to be cytoskeletal in nature. We then test the presumed requirement of SRF-MRTF interactions using a novel $Srf^{al}$ allele carrying mutations that prevent SRF-MRTF-DNA ternary complex formation to circumvent MRTF redundancy and control for possible SRF-independent MRTF activity. These embryos have striking developmental defects that are outwardly similar to *Myocd* mutant mice. Conditional NC mutants reveal an essential role for optimal SRF-MRTF activity in the cardiac crest, whereas the mutation is well-tolerated in the cranial NC. These observations raise the possibility that non-cardiovascular tissue may be able to develop with only minimal SRF-MRTF activity or that SRF can support cytoskeletal gene expression on its own or with other cofactors.

## Results

### Srf^flox/flox; Wnt1-Cre^Tg/+ mice develop a midfacial cleft and bleb, characterized by reduced cytoskeletal gene expression

To establish a phenotypic baseline for embryos lacking *Srf* in NC, we first examined conditional null embryos at E11.5 and E12.5 and assessed their morphology. Consistent with our previous study, a midfacial cleft develops from E10.5 to E11.5, becoming prominent at E11.5 as a failure of the medial nasal process and lateral nasal process (MNP and LNP, respectively) to converge at the midline (*Figure 1B–C*; *Vasudevan and Soriano, 2014*). By E12.5, a fluid-filled bleb develops at the midline, often with hemorrhaging into the midfacial cavity (*Figure 1B*). Embryos turned necrotic starting at E12.5 and did not survive past E13.5.

We sought to better understand the molecular defects that underly this outcome. As *Srf* has been implicated in mediating cell differentiation, we asked whether early craniofacial patterning was affected. However, expression of the differentiation markers *Msx1* (craniofacial mesenchyme), *Alx3* (MNP and LNP mesenchyme, medial mandibular mesenchyme), and *Six3* (ventral forebrain, nasal placode, eye), as well as the markers of patterning centers *Shh* (ventral forebrain, weak oral MNP, and mandibular epithelium), and *Fgf8* (ventral forebrain, oral MNP, and epithelium) were all unaffected at E10.5 as assessed by in situ hybridization, suggesting craniofacial patterning was largely normal at this stage (*Figure 1—figure supplement 1*).

We next sought to identify differentially expressed genes (DEGs) through bulk RNA-sequencing of control and mutant frontonasal prominences (FNP, i.e. MNP+ LNP) and mandibles at E11.5. To confirm the quality of the dataset and suitability of the analysis pipeline, we first compared mandible versus FNP gene expression among all samples and identified differentially expressed transcripts encoding 4084 DEGs (q ≤ 0.05, Wald test), among them known regulators of mandible or FNP identity, such as *Hand2* and *Six3* (*Figure 1—figure supplement 2A-B*). Principal component analysis showed strong separation of the samples by tissue (*Figure 1—figure supplement 2C*). We next identified DEGs in control versus $Srf^{flox/flox}$; $Wnt1-Cre^{Tg/+}$ mandibles and FNPs. Mandibles showed 40 DEGs and FNPs 219 (q ≤ 0.05, Wald test). A joint model including both tissue samples and accounting for tissue-of-origin identified 381 DEGs (*Figure 1D–E*). *Srf* itself was among the top DEGs, confirming efficient conditional deletion in the cranial NC (*Figure 1D–E*, *Figure 1—figure supplement 2D*), but SRF cofactors were not affected (*Figure 1—figure supplement 2E*).

The most differentially expressed genes primarily encoded cytoskeletal genes that were known targets of SRF-MRTF activity, including *Actg1*, *Cnn2*, *Vcl*, *Actb*, and *Cfl1* (*Figure 1D–E*). We subjected a more stringent list of 43 downregulated and 36 upregulated genes with q ≤ 0.05 and Log2FC ≥ 0.25 to gene set enrichment analysis using the online tool Enrichr (*Xie et al., 2021*). Downregulated genes were enriched for cytoskeletal GO terms and SRF-binding motifs (*Figure 1F*), whereas upregulated genes showed little enrichment for either TF motifs or GO terms and may not be direct SRF targets (*Figure 1—figure supplement 2G*). We then used gene set enrichment analysis to compare our results with known targets of SRF, MRTF, and TCF (*Esnault et al., 2014*; *Gualdrini et al., 2016*; *Supplementary file 1*). All three gene lists showed enrichment in our dataset, but the MRTF list was most significantly enriched (*Figure 1G*). Furthermore, limiting this comparison to DEGs with q ≤ 0.05, genes bound by MRTF or MRTF and TCF were more significantly affected (greater fold change) than those bound by TCF alone (*Figure 1H–I*). We also performed these analyses on the individual FNP and Mandible datasets and found similar enrichment for SRF motifs, cytoskeleton-related GO terms, and a stronger enrichment for known MRTF targets than for TCF targets (*Figure 1—figure supplement 2H-I*). The one major difference between the tissues was that the FNP dataset contained a group of uniquely affected genes that enriched for terms related to cholesterol metabolism, but these were not further investigated (*Figure 1—figure supplement 2H*, GO Biological Process).

In summary, our gene expression analysis found that the genes most affected by loss of *Srf* in both mandible and FNP were enriched for cytoskeleton-related established SRF-MRTF targets. These data, coupled with our previous observations that *Pdgfra* interacted genetically with *Srf* in NC and PDGF stimulation promoted SRF-MRTF complex formation, led us to hypothesize that SRF-MRTF interactions would be critical for midfacial development.

## Srf$^{al/al}$ succumb during early organogenesis with cardiovascular defects

In order to test the requirement for SRF-MRTF interactions genetically, we introduced four knock-in point mutations to the αI helix of the SRF DNA-binding domain previously shown to disrupt SRF-MRTF-DNA ternary complex formation while leaving SRF-TCF-DNA complex formation unaffected (*Figure 2—figure supplement 1A-C*; *Hipp et al., 2019*; *Zaromytidou et al., 2006*). Underscoring their importance, we found these residues are conserved in *Srf* orthologs from human to sponge, although they are intriguingly less well-conserved in clades lacking a readily identifiable *Mrtf* ortholog, namely flatworms (*Platyhelminthes*) and Placazoa (*Figure 2—figure supplement 2*). We included an N-terminal 3xFLAG tag and refer to the allele as *Srf$^{al}$*. As a control, we generated a separate *Srf$^{FLAG}$* tagged line without the αI helix mutations.

*Srf$^{FLAG/FLAG}$* mice were viable and fertile, confirming that neither the FLAG tag nor targeting strategy affected development. In contrast, no *Srf$^{al/al}$* mice were found at weaning age (*Table 1*). Because we observed no stillborn or dying neonates, we examined embryos at different stages. *Srf$^{al/al}$* embryos were recovered in Mendelian ratios until E10.5 but were easily identifiable from E9.5 onward due to their obvious morphological differences from control littermates. Mutant embryos were slightly smaller at E9.5 and most had turning defects that ranged in severity from incompletely turned to totally unturned (*Figure 2A*). This was accompanied by a wavy neural tube, as seen in many embryos with a deficiency in mesoderm (*Figure 2A*, middle embryo) and some embryos showed a delay in anterior neural tube closure, indicated by the open midbrain (*Figure 2A*). Mutant embryos also had a missing or hypoplastic second pharyngeal arch (*Figure 2A–B*, asterisks). Additionally, the yolk sac

**Table 1.** Srf$^{al/al}$ embryos are not recovered at weaning.
Expected and recovered numbers of embryos of each genotype at weaning (P21). No homozygous mutant embryos were recovered.

| Genotype | Expected | Observed |
|---|---|---|
| *Srf$^{+/+}$* | 7.75 | 12 |
| *Srf$^{al/+}$* | 15.5 | 19 |
| *Srf$^{al/al}$* | 7.75 | 0 |

$X^2$ Test = 0.0044.

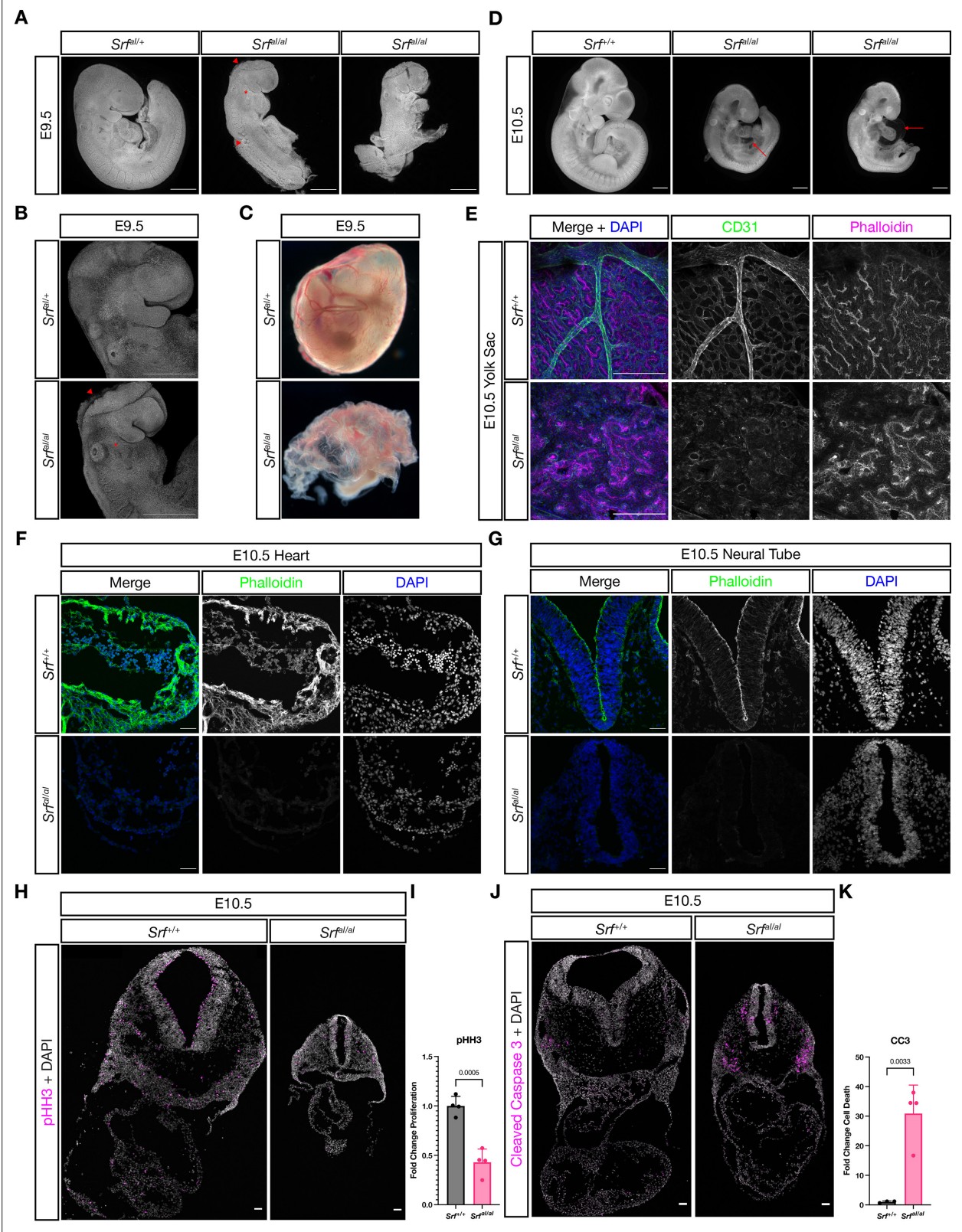

**Figure 2.** Srf^al/al embryos succumb at E10.5 with numerous defects. (**A**) DAPI stained E9.5 embryos of the indicated genotypes show that compared to Srf^al/+ embryos, Srf^al/al embryos are growth retarded, incompletely turned, have short and disorganized tails, a wavy neural tube (red arrowhead, trunk), delayed anterior neural tube closure (red arrowhead, head), and a hypoplastic or missing second pharyngeal arch (red asterisk). Scale bar represents 500 μm. (**B**) Higher magnification confocal images of the first two embryos in (**A**). Scale bar represents 500 μm. (**C**) Brightfield images of E9.5 yolk sacs

*Figure 2 continued on next page*

*Figure 2 continued*

indicate defective vasculogenesis in mutant embryos. (**D**) DAPI-stained E10.5 embryos show more extensive growth retardation, a distended heart tube, and pericardial edema (red arrow). Scale bar represents 500 μm. (**E**) Immunofluorescent staining of E10.5 yolk sacs shows that mutant yolk sacs lack a remodeled vascular plexus or any large vessels. Images are representative of n = 4 embryos of each genotype. Scale bar represents 250 μm. (**F**) Transverse sections through E10.5 embryos at the level of the heart show reduced F-actin intensity via phalloidin staining. Images are representative of n = 4 embryos of each genotype. Scale bar represents 50 μm. (**G**) A similar pattern Is seen in the neural tube. Scale bar represents 50 μm. (**H**) Cell proliferation, indicated through phospho-Histone H3 (Ser10) (pHH3) staining, is reduced in mutant embryos. Scale bar represents 50 μm. (**I**) Quantitation of (**H**), n = 4 each genotype. p = 0.0005, Student's unpaired two-tailed t-test. (**J**) Cell death, revealed through cleaved caspase three staining, is dramatically increased in mutant embryos. Scale bar represents 50 μm. (**K**) Quantitation of (**J**), n = 3 control embryos and n = 4 mutant embryos. p = 0.0033, Student's unpaired two-tailed t-test. Columns are the mean and error bars represent the standard deviation in (**I**) and (**K**).

The online version of this article includes the following source data and figure supplement(s) for figure 2:

**Figure supplement 1.** Targeting strategy and validation for Srf$^{Flag}$and Srf$^{al}$alleles.

**Figure supplement 1—source data 1.** Full gel image of the original scan of the Southern blot used in *Figure 2—figure supplement 1B*.

**Figure supplement 1—source data 2.** Full gel image of the Southern blot used in *Figure 2—figure supplement 1B* after contrast adjustment.

**Figure supplement 2.** SRF αI helix residues are highly conserved, but drift in clades lacking clear MRTF homologues.

showed a crinkled appearance with numerous red blood cells, but no obvious mature blood vessels, indicating the onset of primitive hematopoiesis but not vasculogenesis (*Figure 2C*).

By E10.5, the anterior neural tube had closed but other defects remained or became apparent. Mutant embryos were much smaller than their wild-type or heterozygous littermates (*Figure 2D*). The developing heart tube appeared distended and thin, and most embryos showed pericardial edema (*Figure 2D*, arrows). The overall length of mutant embryos was shorter (*Figure 2D*) and a subset failed to turn, remaining inflected similar to the rightmost embryo in *Figure 2A* (data not shown). Whole-mount immunostaining of the yolk sac with the endothelial marker CD31/PECAM1 revealed that while wild-type littermates had an extensively remodeled capillary plexus, including the presence of larger vessels, mutant yolk sacs had only a crude primitive capillary plexus, despite the presence of CD31-positive cells (*Figure 2E*). A reduction in F-Actin levels throughout mutant embryos, including the heart (*Figure 2F*) and neural tube (*Figure 2G*), was consistent with reduced SRF-MRTF-mediated transcription of cytoskeletal genes. Additionally, mutant embryos showed reduced cell proliferation and significantly increased cell death (*Figure 2H–K*).

While striking, the phenotype of *Srf$^{al/al}$* embryos is less severe than that reported for *Srf* null mutants, which succumb from E6.5-E8.5 and do not induce expression of the mesoderm marker *T* (*Arsenian et al., 1998*). We generated homozygous *Srf$^{-/-}$* embryos and found them to be delayed at E6.5 and E7.5 and were not recovered at later stages (data not shown), verifying the early lethality on our genetic background and thus confirming the difference in severity between the *Srf$^-$* and *Srf$^{al}$* alleles.

## The *Srf$^-$* and *Srf$^{al}$* alleles cause similar defects in the anterior mesodermal lineage

In order to make a second comparison between the *Srf$^{al}$* and *Srf$^-$* alleles, we generated *Srf$^{flox/flox}$*; *Mesp1$^{Cre/+}$* and *Srf$^{al/flox}$*; *Mesp1$^{Cre/+}$* embryos and assessed them at E9.5 and E10.5. *Mesp1-Cre* directs recombination in anterior mesoderm, including cardiac mesoderm. This is a tissue where SRF-MRTF interactions are known to be required, particularly through SRF-Myocardin activity in the developing heart and vascular smooth muscle (*Li et al., 2003*; *Miano et al., 2004*; *Niu et al., 2005*; *Parlakian et al., 2004*). In addition, the wavy neural tube phenotype that we observed in *Srf$^{al/al}$* is often associated with mesoderm deficiency.

Both *Srf$^{flox/flox}$*; *Mesp1$^{Cre/+}$* and *Srf$^{al/flox}$*; *Mesp1$^{Cre/+}$* embryos were inviable after E10.5 and exhibited similar phenotypes. Mutant embryos were small, had turning defects (or arrested prior to or during the turning process), pericardial edema, and hypoplastic hearts. They appeared quite similar to *Srf$^{al/al}$* embryos and phenocopied *Myocd* mutants (*Figure 3A–B*; *Li et al., 2003*). At E9.5, we observed a wavy neural tube in *Srf$^{al/flox}$*; *Mesp1$^{Cre/+}$* embryos, indicating this phenotype is at least partially attributable to defects in mesoderm. Notably, although *Srf$^{al/flox}$*; *Mesp1$^{Cre/+}$* and *Srf$^{flox/flox}$*; *Mesp1$^{Cre/+}$* embryos were broadly similar, *Srf$^{flox/flox}$*; *Mesp1$^{Cre/+}$* embryos were more strongly affected, being reproducibly smaller and completely unturned. Although these embryos were generated from separate crosses, precluding direct comparisons, the observations were consistent across multiple litters. We conclude

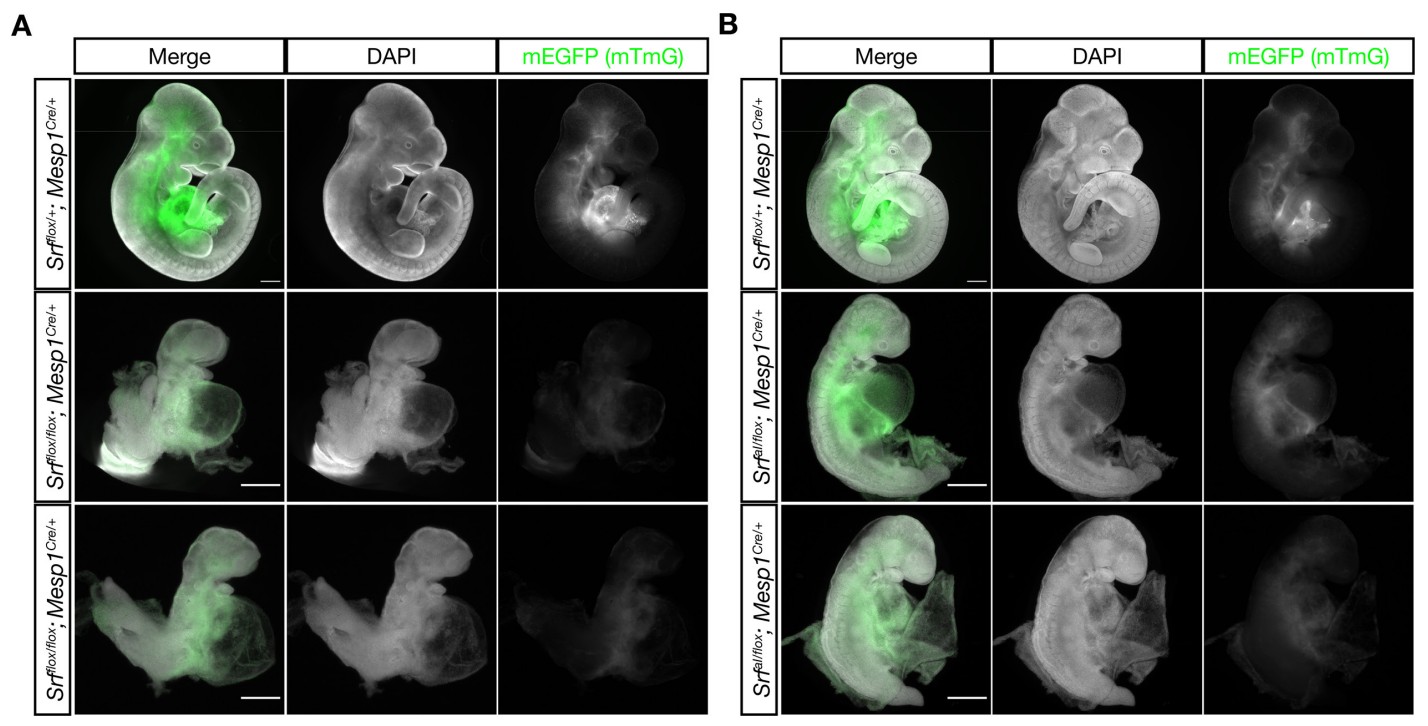

**Figure 3.** The Srf^al and Srf^flox alleles exhibits similar defects in anterior mesoderm. (**A**) E10.5 littermate embryos were stained with DAPI and imaged. Loss of *Srf* in the *Mesp1-Cre* lineage causes embryos to be undersized with pericardial edema, hypoplastic hearts, and turning defects. Phenotype observed in n = 3/3 mutant embryos. (**B**) A similar experiment in which *Srf^al* is the only *Srf* allele expressed in the *Mesp1-Cre* lineage. These embryos appear comparable to the mutant embryos in (**A**), although they are clearly less severely affected as they are slightly larger and partially turned. Phenotype observed in n = 3/3 mutant embryos. Scale bar represents 500 μm in all images. Note the 2 x higher crop in mutant embryos to better illustrate phenotypes.

from this analysis that the *Srf^al* allele is less severe than the *Srf^-* allele, but nevertheless represents a significant curtailment of SRF activity. Moreover, because *Srf^al/flox*; *Mesp1^Cre/+* and *Srf^al/al* embryos are so similar, the *Srf^al/al* phenotype is not a secondary consequence of placental insufficiency, a common cause of cardiovascular and neural phenotypes (*Perez-Garcia et al., 2018*), as *Mesp1^Cre/+* labels anterior embryonic and extraembryonic (i.e. yolk sac) mesoderm, but not the trophectoderm-derived placenta (*Saga et al., 1999*).

## Srf^al/flox; Wnt1-Cre^Tg/+ embryos do not display craniofacial defects at E13.5

We next asked whether *Srf^al/flox*; *Wnt1-Cre^Tg/+* embryos would display similar defects to *Srf^flox/flox*; *Wnt1-Cre^Tg/+* embryos, as we expected. Surprisingly, these embryos appeared completely normal at E13.5 (*Figure 4A*), when *Srf^flox/flox*; *Wnt1-Cre^Tg/+* embryos are already dying and display obvious craniofacial abnormalities.

We tested whether the *Srf^al* allele was behaving as expected in the NC lineage using MEPM cells cultured from E13.5 *Srf^al/flox*; *Wnt1-Cre^Tg/+*; *ROSA26^TdT/+* (mutant) and *Srf^flox/+*; *Wnt1-Cre^Tg/+*; *ROSA26^TdT/+* (control) palatal shelves (*Figure 4B*). We assessed the expression of genes preferentially regulated by SRF-TCF activity, such as the IEGs, and those regulated by SRF-MRTF activity, namely cytoskeletal genes. At the protein level, immunofluorescent staining of MEPM cells for F-actin and smooth muscle actin (SMA) showed reduced intensity in mutant cell lines compared to control lines (*Figure 4C*). We also assessed gene expression at the mRNA level by qPCR in starved and serum-stimulated lines. While we noted no significant changes in the expression of the IEGs *Egr1* and *Fos* (*Figure 4D*), levels of *Tagln* (SM22) and *Acta2* (the gene encoding SMA) were significantly downregulated in both conditions, and we noted a downward trend in *Vcl* expression (*Figure 4D*). *Srf* itself was also significantly downregulated, likely due to autoregulation via the several CArG elements at the *Srf* locus

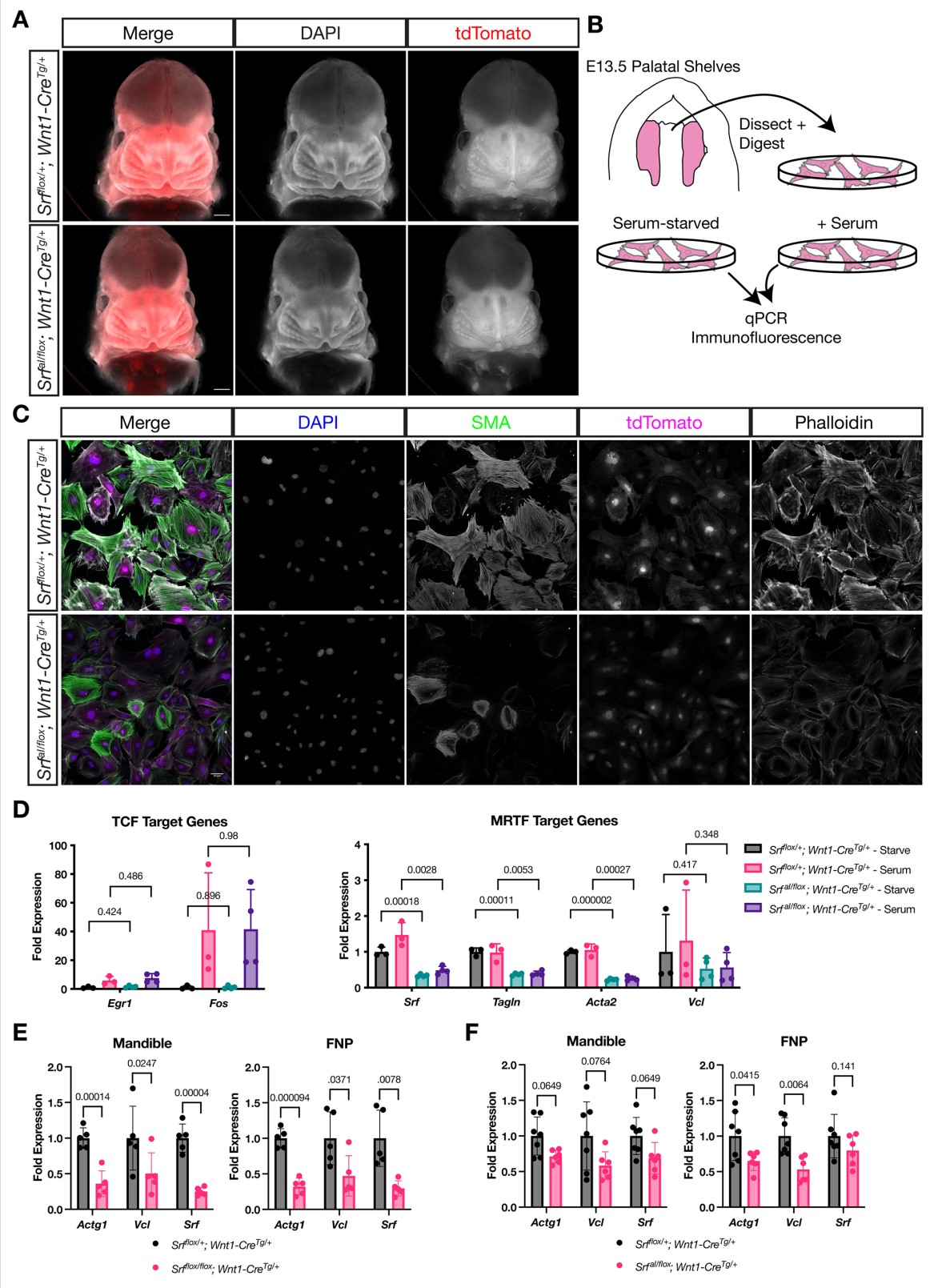

**Figure 4.** NC Srf[al]conditional mutants are normal at E13.5. (**A**) DAPI stained E13.5 littermate embryos carrying a *ROSA26*[TdT/+] Cre reporter show no apparent craniofacial defects in conditional mutants. Scale bar represents 500 µm. (**B**) Diagram illustrating the culture of MEPM cells. (**C**) Immunofluorescent staining of passage 2 MEPM cells shows reduced F-actin and SMA fluorescence in mutant cells compared to cells from heterozygous littermate control embryos. Scale bar represents 50 µm. (**D**) RT-qPCR from serum-starved and serum-stimulated MEPM cells indicates

*Figure 4 continued on next page*

*Figure 4 continued*

no difference in IEG expression (*Egr1, Fos*) but a significant defect in *Srf, Tagln,,* and *Acta2* expression and a downward trend in *Vcl* expression. Values are fold expression of control starved cells. N = 3 control lines and n = 4 mutant lines. (**F**) RT-qPCR from E11.5 FNPs and mandibles shows a significant reduction in *Actg1, Vcl,* and *Srf* expression in *Srf* NC conditional knockouts compared to control conditional heterozygous littermates. N = 5 each genotype. (**G**) Assaying the same genes and tissues as (**F**) using *Srf*^al^ NC conditional mutants shows a downward trend or significant reduction approximately half as large as (**F**). N = 7 controls and n = 6 mutants. For (**E–G**) significance was determined by Student's unpaired t-test with two-stage step-up correction (Benjamini, Krieger, and Yekutieli) for multiple comparisons. q-values are indicated on the graphs. Columns are the mean and error bars represent the standard deviation.

The online version of this article includes the following figure supplement(s) for figure 4:

**Figure supplement 1.** MRTF-A translocates to the nucleus normally in response to serum stimulation in mutant MEPM cells.

(*Figure 4D*, *Figure 2—figure supplement 1A*). We confirmed that MRTF-A translocated to the nucleus in response to serum in control and mutant cell lines, indicating that the regulated localization of MRTF-A is not affected but that SRF^al^ fails to productively interact with MRTF-A despite its nuclear localization, thereby affecting transcription of MRTF targets (*Figure 4—figure supplement 1A-B*).

Since we saw striking changes in these cells in culture but no phenotype up to this stage in the embryo, we sought to compare gene expression differences in *Srf*^al/flox^; *Wnt1-Cre*^Tg/+^ and *Srf*^flox/flox^; *Wnt1-Cre*^Tg/+^ embryos. We assessed two of the most strongly affected DEGs from our RNA-Seq, *Actg1* and *Vcl*, and *Srf* itself in both models. In E11.5 *Srf*^flox/flox^; *Wnt1-Cre*^Tg/+^ facial prominences, we found a strong reduction in expression of all three genes, confirming our RNA-Seq data (*Figure 4E*). The same genes were downregulated in *Srf*^al/flox^; *Wnt1-Cre*^Tg/+^ prominences, but to a lesser degree than the conditional knockouts (*Figure 4F*), potentially accounting for the lack of phenotype at this stage.

Because *Srf* interacts genetically with *Pdgfra* in the NC and because SRF-MRTF transcriptional targets were suggested to be of particular importance downstream of PDGFRA signaling, we reasoned that the *Srf*^al^ allele might also interact genetically with *Pdgfra* in this tissue (*Vasudevan and Soriano, 2014*). To test this possibility, we generated *Srf*^al/+^; *Pdgfra*^H2B-EGFP/+^; *Wnt1-Cre*^Tg/+^ triple heterozygous male mice and crossed them with *Srf*^flox/flox^; *ROSA26*^TdT/TdT^ mice but did not observe facial clefting in embryos of any genotype (data not shown).

In summary, cells from *Srf*^al/flox^; *Wnt1-Cre*^Tg/+^ embryos show the expected changes in gene expression, yet the embryos themselves show no outward signs of the severe craniofacial phenotypes observed in *Srf*^flox/flox^; *Wnt1-Cre*^Tg/+^ embryos at this stage and have milder defects in gene expression.

## *Srf*^al/flox^; Wnt1-Cre^Tg/+^ mice succumb in the early postnatal period with outflow tract defects

*Srf*^al/flox^; *Wnt1-Cre*^Tg/+^ embryos survived until birth, after which they died within the first 2 days of life with visible cyanosis (*Figure 5A*). We examined E18.5 skeletal preparations for defects in patterning or ossification that may arise after E13.5. Mutant skulls were smaller than in control littermates, but the craniofacial skeleton was patterned normally indicating a developmental delay at this timepoint (*Figure 5—figure supplement 1*). To determine the underlying cause of cyanosis, we examined the cardiac outflow tract at P0 as the smooth muscle in this region is NC-derived and responsible for proper remodeling of the aortic arch vessels during development. We found a highly penetrant (9/14) patent ductus arteriosus (PDA) exclusively in *Srf*^al/flox^; *Wnt1-Cre*^Tg/+^ neonates (*Figure 5B–C*). In this condition, the embryonic shunt from the pulmonary artery to the aorta, the ductus arteriosus, fails to close after birth, making circulation to the lungs inefficient and likely explaining the postnatal cyanosis. We also noted one instance of aberrant right subclavian artery, in which the right subclavian artery originates from the descending aorta instead of the brachiocephalic artery, which only supplies the right common carotid artery in this condition. We also inspected P0 mice from a similar cross on a *Pdgfra*^H2B-EGFP/+^ background to assess whether heterozygosity for *Pdgfra* would exacerbate phenotypes at this stage, but neonates were recovered in the expected Mendelian ratios with similar outflow tract defects (*Figure 5—figure supplement 2A-B*). Two *Srf*^al/flox^; *Wnt1-Cre*^Tg/+^; *Pdgfra*^+/+^ from this set of crosses displayed a more severe outflow tract defect: right aortic arch with mirror image branching (2/9; *Figure 5C*, far right).

Sections through *Srf*^al/flox^; *Wnt1-Cre*^Tg/+^ and control *Srf*^flox/+^; *Wnt1-Cre*^Tg/+^ hearts confirmed the macroscopically observed PDAs and also revealed one instance of ventricular septal defect (VSD) with persistent truncus arteriosus (PTA), a failure of the truncus arteriosus to fully septate into the aorta and

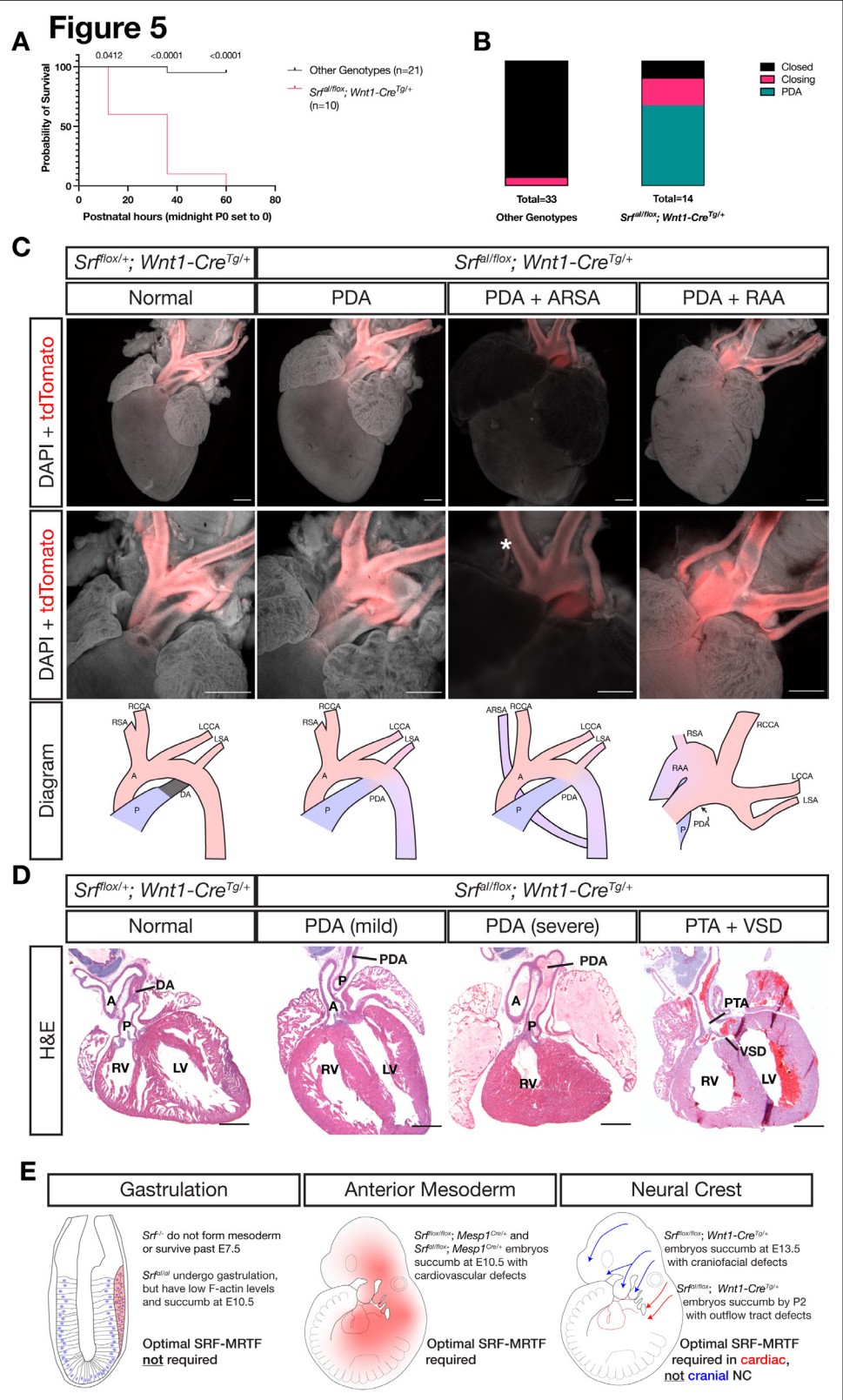

**Figure 5.** NC Srf$^{al}$ conditional mutants succumb postnatally with outflow tract defects. (**A**) Kaplan-Meyer survival curve for neonatal Srf$^{al/flox}$; Wnt1-Cre$^{Tg/+}$ (Mutant) compared to littermates of all other genotypes (Control). Significance was computed at each timepoint using a Mantel-Cox log-rank test. P0 p = 0.0142, P1 p < 0.0001, P2 p < 0.0001. (**B**) Stacked columns showing the distribution of PDA-related phenotypes in Srf$^{al/flox}$; Wnt1-Cre$^{Tg/+}$

*Figure 5 continued on next page*

*Figure 5 continued*

neonates compared to littermates of all other genotypes. (**C**) DAPI-stained postnatal day 0 (P0) hearts carrying a *ROSA26^TdT/+^* Cre lineage reporter showing the entire heart (top row) and the outflow tract region (bottom row). Examples of mutant phenotypes such as PDA, ARSA, and RAA. An asterisk indicates where the missing right subclavian artery should be. Note the ARSA mouse had succumbed prior to dissection and the image is dimmer due to the presence of clotted blood. The outflow tract defects are schematized below. Scale bar represents 500 µm. (**D**) Hematoxylin and eosin-stained frontal sections through P0 hearts showing mild to severe PDA and an example of VSD and PTA in mutants. Scale bar represents 500 µm. (**E**) Summary of our results, showing the requirements for SRF versus SRF^al^ in different tissues and timepoints. A, aorta; ARSA, aberrant right subclavian artery; DA, ductus arteriosus; LCA, left common carotid artery; LSA, left subclavian artery; LV, left ventricle; P, pulmonary artery; PDA, patent ductus arteriosus; PTA, persistent truncus arteriosus; RAA, right aortic arch with mirror image branching; RCA, right common carotid artery; RSA, right subclavian artery.

The online version of this article includes the following figure supplement(s) for figure 5:

**Figure supplement 1.** E18.5 conditional mutant skulls are delayed but correctly patterned.

**Figure supplement 2.** Pdgfra *and* Srf^al^ do not interact genetically in NC.

pulmonary artery, out of ten mutant hearts examined (*Figure 5D*). Mutant embryos that survived the first several hours of life had a milk spot, indicating they did not have pronounced defects in olfaction or the craniofacial bones, nerves, and muscles required for suckling. Thus, while the *Srf^al^* allele is surprisingly well-tolerated in the cranial NC lineage throughout most of development, it is required in the cardiac NC-derived smooth muscle of the cardiac outflow tract to support postnatal life, highlighting a critical role for SRF-MRTF interactions in this particular NC lineage (*Figure 5E*).

## Discussion

SRF is a ubiquitously expressed transcription factor whose transcriptional output is strongly influenced by its cofactors, the TCFs and MRTFs. These cofactors, in turn, are regulated both by specific expression patterns (e.g. *Myocd* specifically in muscle) and by signaling pathways, such as ERK1/2 and PI3K (*Posern and Treisman, 2006*; *Figure 1A*). We sought to better understand the relationship between SRF, its cofactors, and mutant phenotypes in both a general and a tissue-specific manner.

Our previous study demonstrated a requirement for *Srf* in the NC lineage (*Vasudevan and Soriano, 2014*), so we first investigated how loss of SRF in this tissue affected gene expression. We found normal early NC patterning but misexpression of cytoskeleton-related genes in *Srf^flox/flox^*; *Wnt1-Cre^Tg/+^* embryos (*Figure 1C–F*, *Figure 1—figure supplements 1 and 2E*). Comparing the DEGs with known SRF and cofactor targets showed particular enrichment for MRTF targets in our datasets (*Figure 1G–H*, *Figure 1—figure supplement 2F-G*), consistent with SRF's well-known function regulating cytoskeletal genes and the dominant role of MRTF cofactors in the serum response (*Esnault et al., 2014*). To test the supposition that SRF-MRTF activity would be the main driver of NC SRF activity, we made a new *Srf^al^* mouse model harboring point mutations that specifically disrupt SRF-MRTF-DNA complex formation (*Zaromytidou et al., 2006*).

Homozygous *Srf^al/al^* embryos died at E10.5 with defects in the yolk sac vasculature, heart, and neural tube and exhibited reduced F-actin levels (*Figure 2A–F*). Both the timing of lethality and gross appearance of the embryos strongly resemble *Myocd^-/-^* mutant mice (*Li et al., 2003*). The phenotype is much less severe than *Srf^-/-^* embryos, however, which die around E7.5 and fail to induce mesoderm (*Arsenian et al., 1998*; *Niu et al., 2005*). Expression of the *Srf^al^* allele specifically in the *Mesp1^Cre^* mesodermal lineage also resulted in lethality at E10.5 and defects almost as severe as complete loss of *Srf* in this same lineage (*Figure 3*). The similarity of the two alleles in the *Mesp1^Cre^* lineage compared against their strikingly different phenotypes in gastrula-stage embryos suggested there may be time or tissue specific requirements for SRF-MRTF activity.

We went on to test the *Srf^al^* allele in the NC lineage, where we presumed it would be critically important. Similar to the early embryo, expression of the *Srf^al^* allele was well tolerated in NC and we found no facial cleft or bleb in our *Srf^al/flox^*; *Wnt1-Cre^Tg/+^* mice, as in *Srf^flox/flox^*; *Wnt1-Cre^Tg/+^* mice (*Figure 4A*). The allele also failed to interact genetically with *Pdgfra* in this lineage, suggesting PDGF signaling may rely on MRTF-independent SRF activity (*Figure 5—figure supplement 2A-B*). Nevertheless, this mutation in NC does have a profound developmental effect as *Srf^al/flox^*; *Wnt1-Cre^Tg/+^* mice

completed gestation but died by postnatal day 2 with visible cyanosis (*Figure 5A*). Examination of the cardiac outflow tract revealed numerous defects including PDA, right aortic arch, and one instance of VSD with PTA (*Figure 5C–D*). This result is reminiscent of mice with a hypomorphic gene trap mutation in *Mrtfb* and mice carrying a conditional deletion of *Myocd* in the NC (*Huang et al., 2008*; *Li et al., 2005b*), both of which result in early postnatal lethality due to defects in outflow tract development. Similarly, NC conditional *Srf* mutants on a genetic background that permitted later development exhibit outflow tract defects at E16.5 (*Newbern et al., 2008*). These studies and our own results together highlight a critical role for SRF-MRTF interactions in cardiac NC development. A summary of the tissue-specific sensitivities we found to loss of *Srf* or *Srf*$^{al}$ expression is depicted in *Figure 5E*.

We considered three possible explanations for the tolerance of the *Srf*$^{al}$ allele in NC and early embryo, which are not mutually exclusive: (i) that TCFs are more important than previously recognized or are redundant with MRTFs in some respects, (ii) that different tissues may have different thresholds for SRF-MRTF activity and that our point mutant may interact differently with different MRTF family members, and (iii) that SRF may utilize cofactors other than MRTFs and TCFs or have cofactor-independent (i.e. basal) activity sufficient to support development in some contexts.

The first possibility is that TCF factors play a more important role than previously thought or can act redundantly with MRTFs. Because TCF triple mutant embryos survive until E14.5 without obvious morphological defects (*Costello et al., 2010*), we find it unlikely that TCF-specific activity could explain the tolerance of *Srf*$^{al}$ during gastrulation or in NC. However, it has been shown that some SRF targets can be bound and regulated by both MRTF and TCF factors (*Esnault et al., 2014*), raising the possibility of cofactor redundancy. Yet, most studies to date indicate that rather than acting redundantly, MRTFs and TCFs mediate distinct and opposing phenotypic outcomes, contractility and proliferation, respectively (*Gualdrini et al., 2016*; *Wang et al., 2004*). Indeed, a potential result of this competition is that the consequences of suppressing SRF-MRTF interactions could vary according to the level of TCF expression in a given cell type. While we cannot rule out the possibility that TCFs are the primary SRF cofactors in NC or function redundantly to MRTFs without further genetic experiments, this explanation is difficult to reconcile with the existing literature. Although there is no known *Srf* allele that blocks TCF binding while leaving MRTF binding unperturbed, the V194E mutation (V189E in mouse) disrupts SRF interaction with both cofactors (*Ling et al., 1998*). Generating a mouse model of this allele would shed considerable light on the question of TCF/MRTF redundancy.

A second explanation is that non-muscle lineages may be able to function with minimal, but not zero, SRF-MRTF activity. It is possible that the *Srf*$^{al}$ allele substantially impairs but does not eliminate SRF-MRTF-DNA complex formation and functions as a hypomorph. In vitro gel shift assays for SRF-MRTF-DNA complex formation using purified components and single molecule imaging of SRF in cells both indicate a substantial disruption of SRF-MRTF activity for this allele, but the gel shift experiments that first characterized these mutations detected 5–10% residual complex formation (*Hipp et al., 2019*; *Zaromytidou et al., 2006*). Our own data demonstrate that homozygous *Srf*$^{al/al}$ embryos grossly phenocopy *Myocd*$^{-/-}$ mutants (*Figure 2*) and *Srf*$^{al/flox}$; *Mesp1-Cre*$^{Tg/+}$ and *Srf*$^{flox/flox}$; *Mesp1-Cre*$^{Tg/+}$ embryos were similar, though not identical (*Li et al., 2003*). On the other hand, double conditional mutants for *Mrtfa/Mrtfb* largely *do* phenocopy *Srf* conditional mutants in several tissues (*Cenik et al., 2016*; *Guo et al., 2018*; *Li et al., 2005a*; *Trembley et al., 2015*). Therefore, we may be observing differing dosage requirements for SRF-MRTF activity in distinct tissues. Muscle lineages, such as cardiovascular cells affected by *Mesp1*$^{Cre}$ and NC-derived outflow tract smooth muscle affected by *Wnt1-Cre,* may need optimal SRF-MRTF output and are therefore strongly affected by the *Srf*$^{al}$ allele. Non-muscle lineages such as the cranial NC may survive and develop properly with only residual SRF-MRTF transcription. The *Srf*$^{al}$ allele might more strongly suppress Myocardin compared to MRTFA/B, which could also contribute to the tissue-specific requirements we observe. Two predictions of this threshold model are that conditional ablation of *Mrtfa/Mrtfb* in NC would phenocopy conditional loss of *Srf* and that *Mrtfa/*Mrtfb double mutant embryos should succumb around E7.5 as *Srf* null embryos do. Conversely, non-muscle lineages where conditional ablation of *Mrtfa/Mrtfb* yields *Srf*-like phenotypes, such as podocytes and epicardium, should be indifferent to the *Srf*$^{al}$ mutations.

The issue of SRF tissue-specific dosage effects may have relevance to human disease. A recent study performed targeted sequencing of *SRF* in nonsyndromic conotruncal heart defect patients and identified two novel mutations with reduced transcriptional output, one from a patient with VSD and the other with Tetralogy of Fallot with right aortic arch (*Mengmeng et al., 2020*). Thus, tuning of SRF

output may modulate disease in a tissue-specific manner. Along these lines, mutations in *MYOCD* cause congenital megabladder and associated cardiovascular phenotypes such as PTA and VSD in humans, but monoallelic mutations affect only males whereas biallelic mutations affect both sexes (*Houweling et al., 2019*). Furthermore, heterozygosity for *FLNA*, a gene we found strongly affected by loss of *Srf* in mouse NC, causes the human disease Periventricular Heterotopia I and affected females present with PDA, whereas hemizygous males die during gestation (*Fox et al., 1998*). Intriguingly, NC-specific conditional knockout of *Flna* causes perinatal lethality with outflow tract defects in mice (*Feng et al., 2006*). It would be interesting to further explore the notion of tissue-specific thresholds for SRF-cofactor complexes in future studies.

A third explanation is that SRF functions with additional factors that regulate gene expression independent of or redundantly with MRTFs (and/or TCFs) or has sufficient basal activity to support development in some tissues. Although MRTFs and TCFs are the most well-studied SRF cofactors, many other TFs have been shown to interact with SRF, including but not limited to Homeodomain proteins (*Chen et al., 1996*; *Chen et al., 2002*; *Grueneberg et al., 1992*; *Shin et al., 2002*), GATA factors (*Belaguli et al., 2000*; *Morin et al., 2001*), and Forkhead-family transcription factors (*Freddie et al., 2007*; *Liu et al., 2005*), as well as the Initiator-binding protein TFII-I (*Grueneberg et al., 1997*; *Kim et al., 1998*). Many of these studies were performed in muscle cells and it is unclear which cofactors might act independently of MRTFs. However, several of these genes or their orthologues are expressed in the cranial NC at E10.5 and E11.5, when cleft formation begins in $Srf^{flox/flox}$; $Wnt1$-$Cre^{Tg/+}$ embryos (*Minoux et al., 2017*). One candidate is the homeodomain protein PRRX1/PHOX1/MHOX, which was shown to form complexes with SRF mediated by TFII-I (*Grueneberg et al., 1997*). Double mutants for *Prrx1* and its orthologue *Prrx2* have defects of the craniofacial skeleton, aortic arch arteries, and ductus arteriosus (*Bergwerff et al., 2000*; *Lu et al., 1999*). *Gtf2i* mutants (the gene encoding TFII-I) rarely survive past E10.5 but can exhibit a facial cleft, hemorrhaging, and hypoplastic pharyngeal arches (*Enkhmandakh et al., 2009*). Whether SRF mediates transcription on its own, independently of MRTFs and TCFs, or perhaps using additional tissue-specific cofactors, would be exciting to determine.

In conclusion, we found that the primary transcriptional consequence of losing *Srf* in NC was a defect in actin cytoskeleton-related gene expression. Using a novel $Srf^{al}$ allele to perturb SRF's interactions with MRTFs, the primary cofactors regulating the cytoskeletal transcription program, we uncovered a crucial role for SRF-MRTF activity in the cardiac NC, but surprisingly found the mutation well-tolerated in the cranial NC. Further study will be necessary to determine the relevant SRF-cofactor ensembles in different developmental contexts.

# Materials and methods

## Key resources table

| Reagent type (species) or resource | Designation | Source or reference | Identifiers | Additional information |
|---|---|---|---|---|
| Gene (*Mus musculus*) | *Srf* | MGI, Ensembl, UniProt | Srf, ENSMUSG 00000015605, Q9JM73 | |
| Strain, strain background (*Mus musculus*) | 129S4/SvJaeJ | IMSR, Jackson Labs | 009104 | All the subsequent genetic reagents were made on or backcrossed to this strain |
| Genetic reagent (*Mus musculus*) | $Srf^{FLAG}$ | This paper | To be submitted to Jackson Labs | 3x-FLAG tag knocked in to SRF's N-terminus |
| Genetic reagent (*Mus musculus*) | $Srf^{al}$ | This paper | To be submitted to Jackson Labs | Alpha-I helix mutations knocked into SRF and identical 3x-FLAG to above |
| Genetic reagent (*Mus musculus*) | Wnt1-Cre | IMSR, Jackson Labs | $H2az2^{Tg(wnt1-cre)11Rth}$ RID:IMSR_JAX:003829 | |
| Genetic reagent (*Mus musculus*) | $Mesp1^{Cre}$ | IMSR, Jackson Labs | $Mesp1^{tm2(cre)Ysa}$ RID:IMSR_HAR:3358 | |

*Continued on next page*

*Continued*

| Reagent type (species) or resource | Designation | Source or reference | Identifiers | Additional information |
|---|---|---|---|---|
| Genetic reagent (*Mus musculus*) | Srf^flox^ | IMSR, Jackson Labs | Srf^tm1Rmn^ RRID:IMSR_JAX:006658 | |
| Genetic reagent (*Mus musculus*) | MORE-Cre | IMSR, Jackson Labs | Meox2^tm1(cre)Sor^ RRID:IMSR_JAX:003755 | |
| Genetic reagent (*Mus musculus*) | R26R^TdT^ | IMSR, Jackson Labs | Gt(ROSA) 26Sor^tm14(CAG-tdTomato)Hze^ RRID:IMSR_JAX:007914 | |
| Genetic reagent (*Mus musculus*) | R26R^mTmG^ | IMSR, Jackson Labs | Gt(ROSA) 26Sor^tm4(ACTB-tdTomato,-EGFP)Luo^ RRID:IMSR_JAX:007676 | |
| Cell line (*Mus musculus*) | MEPM | This paper | | Primary cell line derived in lab. Used and extinguished by passage 2. |
| Other | DMEM High Glucose | Invitrogen | 11965118 | |
| Other | Penicillin-Streptomycin (10,000 U/mL) | Gibco | 15140122 | 100 x stock used at 0.5 x |
| Other | L-Glutamine (200 mM) | Gibco | 2 5030081 | 100 x stock used at 1 x |
| Other | Characterized Fetal Bovine Serum, CA Origin | HyClone | SH30396.03 | Lot AC10235406 |
| Antibody | Rat anti-CD31 (rat monoclonal) | BD Biosciences | BD Biosciences Cat# 553370, RRID:AB_394816 | IF(1:50) |
| Antibody | anti-Cleaved Caspase 3 (rabbit monoclonal) | Cell Signaling | Cell Signaling Technology Cat# 9665, RRID:AB_2069872 | IF(1:400) |
| Antibody | anti phospho-Histone H3 (Ser10) (rabbit polyclonal) | Millipore | Millipore Cat# 06–570, RRID:AB_310177 | IF(1:500) |
| Antibody | anti-MKL1 (rabbit polyclonal) | Proteintech | Proteintech Cat# 21166–1-AP, RRID:AB_2878822 | IF(1:100) |
| Antibody | anti-SMA (rabbit monoclonal) | Cell Signaling | Cell Signaling Technology Cat# 19245, RRID:AB_2734735 | IF(1:200) |
| Antibody | Goat anti-Mouse IgG (H + L) Highly Cross-Adsorbed Secondary Antibody, Alexa Fluor Plus 488 | Thermo Fisher | Thermo Fisher Scientific Cat# A32723, RRID:AB_2633275 | IF(1:500) |
| Antibody | Goat anti-Rabbit IgG (H + L) Highly Cross-Adsorbed Secondary Antibody, Alexa Fluor Plus 555 | Thermo Fisher | Thermo Fisher Scientific Cat# A32732, RRID:AB_2633281 | IF(1:500) |
| Other | Phalloidin-Alexa Fluor 647 | Invitrogen | Cat A22287 | IF(1:400) |
| Recombinant DNA reagent | Alx3 (plasmid, in situ probe) | This paper | | doi.10.5061/ dryad.mgqnk9916 |
| Recombinant DNA reagent | Fgf8 (plasmid, in situ probe) | *Crossley and Martin, 1995* | pBS-SK-Fgf8 | doi.10.5061/ dryad.mgqnk9916 |
| Recombinant DNA reagent | Msx1 (plasmid, in situ probe) | *Hong and Krauss, 2012* | | doi.10.5061/ dryad.mgqnk9916 |
| Recombinant DNA reagent | Shh (plasmid, in situ probe) | This paper | | doi.10.5061/ dryad.mgqnk9916 |
| Recombinant DNA reagent | Six3 (plasmid, in situ probe) | This paper | | doi.10.5061/ dryad.mgqnk9916 |
| Sequence-based reagent | Srf_flox_F | *Holtz and Misra, 2008* | genotyping primers | TGCTTACTGG AAAGCTCATGG |
| Sequence-based reagent | Srf_flox_R | *Holtz and Misra, 2008* | genotyping primers | TGCTGGTTTG GCATCAACT |

*Continued on next page*

*Continued*

| Reagent type (species) or resource | Designation | Source or reference | Identifiers | Additional information |
|---|---|---|---|---|
| Sequence-based reagent | Srf_null_R | This paper | genotyping primers | CTAACCCTGC CTGTCCTTCA Use with Srf_flox_F |
| Sequence-based reagent | Srf_flag_F | This paper | genotyping primers | GATGAACGA TGTGACCTCGC |
| Sequence-based reagent | Srf_flag_R | This paper | genotyping primers | AGGGAGGA GCCAACTCCTTA |
| Sequence-based reagent | aR4 | *Hamilton et al., 2003* | genotyping primers | CCCTTGTGG TCATGCCAAAC For Pdgfra$^{EGFP}$ |
| Sequence-based reagent | aR5 | *Hamilton et al., 2003* | genotyping primers | GCTTTTGCC TCCATTA CACTGG For Pdgfra$^{EGFP}$ |
| Sequence-based reagent | lox | *Hamilton et al., 2003* | genotyping primers | ACGAAGTTAT TAGGTCC CTCGAC For Pdgfra$^{EGFP}$ |
| Sequence-based reagent | Cre_800 | This paper | genotyping primers | GCTGCCACGAC CAAGTGACA GCAATG |
| Sequence-based reagent | Cre_1200 | This paper | genotyping primers | GTAGTTATTC GGATCATCAG CTACAC |
| Sequence-based reagent | morefor | *Tallquist and Soriano, 2000* | genotyping primers | GGGACCACC TTCTTTTGGCTTC |
| Sequence-based reagent | morerev | *Tallquist and Soriano, 2000* | genotyping primers | AAGATGTGGAG AGTTCGGGGTAG |
| Sequence-based reagent | morecre | *Tallquist and Soriano, 2000* | genotyping primers | CCAGATCCTC CTCAGAA ATCAGC |
| Sequence-based reagent | R26mTmG_F | *Muzumdar et al., 2007* | genotyping primers | CTCTGCTGC CTCCTGGCTTCT |
| Sequence-based reagent | R26mTmG_wt_R | *Muzumdar et al., 2007* | genotyping primers | CGAGGCGG ATCACAA GCAATA |
| Sequence-based reagent | R26mTmG_mut_R | *Muzumdar et al., 2007* | genotyping primers | TCAATGGGCG GGGGTCGTT |
| Sequence-based reagent | R26Tdt_wt_F | *Madisen et al., 2010* | genotyping primers | AAGGGAGCT GCAGTGGAGTA |
| Sequence-based reagent | R26Tdt_wt_R | *Madisen et al., 2010* | genotyping primers | CCGAAAATC TGTGGGAAGTC |
| Sequence-based reagent | R26Tdt_mut_F | *Madisen et al., 2010* | genotyping primers | GGCATTAAAGC AGCGTATCC |
| Sequence-based reagent | R26Tdt_mut_R | *Madisen et al., 2010* | genotyping primers | CTGTTCCTGT ACGGCATGG |
| Sequence-based reagent | Acta2_qPCR_F | This paper | qPCR primers | GGCACCACT GAACCCTAAGG |
| Sequence-based reagent | Acta2_qPCR_R | This paper | qPCR primers | ACAATACCAG TTGTAC GTCCAGA |
| Sequence-based reagent | Actg1_qPCR_F | This paper | qPCR primers | ATTGTCAATG ACGAGTGCGG |
| Sequence-based reagent | Actg1_qPCR_R | This paper | qPCR primers | CTTACACTGC GCTTCTTGCC |
| Sequence-based reagent | Egr1_qPCR_F | This paper | qPCR primers | TGGGATAACTC GTCTCCACC |
| Sequence-based reagent | Egr1_qPCR_R | This paper | qPCR primers | GAGCGAACAA CCCTATGAGC |

*Continued on next page*

*Continued*

| Reagent type (species) or resource | Designation | Source or reference | Identifiers | Additional information |
|---|---|---|---|---|
| Sequence-based reagent | Fos_qPCR_F | This paper | qPCR primers | TCCTACTACCAT TCCCCAGC |
| Sequence-based reagent | Fos_qPCR_R | This paper | qPCR primers | TGGCACTAGAG ACGGACAGA |
| Sequence-based reagent | Hprt_qPCR_F | This paper | qPCR primers | TCCTCCTCAG ACCGCTTTT |
| Sequence-based reagent | Hprt_qPCR_R | This paper | qPCR primers | CATAACCTGG TTCATCATCGC |
| Sequence-based reagent | Srf_qPCR_F | This paper | qPCR primers | GTGCCACTGG CTTTGAAGA |
| Sequence-based reagent | Srf_qPCR_R | This paper | qPCR primers | GCAGGTTGGT GACTGTGAAT |
| Sequence-based reagent | Tagln_qPCR_F | This paper | qPCR primers | GACTGCACTTC TCGGCTCAT |
| Sequence-based reagent | Tagln_qPCR_R | This paper | qPCR primers | CCGAAGCTAC TCTCCTTCCA |
| Sequence-based reagent | Vcl_qPCR_F | This paper | qPCR primers | TCTGATCCT CAGTGG TCTGAAC |
| Sequence-based reagent | Vcl_qPCR_R | This paper | qPCR primers | AAAGCCATTC CTGACCTCAC |
| Other | BM-Purple | Roche | Cat. #11442074001 | |
| Commercial assay or kit | Luna Universal qPCR Master Mix | New England Biolabs | Cat. #M3003L | |
| Commercial assay or kit | NEBuilder HiFi DNA Assembly Master Mix | New England Biolabs | Cat. #E2621S | |
| Commercial assay or kit | RNeasy Plus Mini Kit | Qiagen | Cat. #74,134 | |
| Chemical compound, drug | Gibco Geneticin (G-418) | Thermo FIsher | Cat. #11811031 | |
| Software, algorithm | Salmon | Salmon | Salmon, RRID:SCR_017036 | |
| Software, algorithm | Wasabi | Wasabi | | https://github.com/COMBINE-lab/wasabi; *Patro, 2019* |
| Software, algorithm | Sleuth | Sleuth | sleuth, RRID:SCR_016883 | |
| Software, algorithm | GraphPad Prism | GraphPad Prism | GraphPad Prism, RRID:SCR_002798 | |
| Other | DAPI stain | Sigma | Cat. #D9542-10mg | (1–5 µg/mL) |

## Animal husbandry

All animal experimentation was conducted according to protocols approved by the Institutional Animal Care and Use Committee of the Icahn School of Medicine at Mount Sinai (LA11-00243). Mice were kept in a dedicated animal vivarium with veterinarian support. They were housed on a 13 hr-11hr light-dark cycle and had access to food and water ad libitum.

## Mouse models

The following previously described mouse lines were used: *H2az2*[Tg(wnt1-cre)11Rth] referred to as *Wnt1-Cre* (*Danielian et al., 1998*), *Mesp1*[tm2(cre)Ysa] referred to as *Mesp1*[Cre] (*Saga et al., 1999*), *Srf*[m1Rmn] referred to as *Srf*[flox] (*Miano et al., 2004*), *Meox2*[tm1(cre)Sor] referred to as *MORE-Cre* (*Tallquist and Soriano, 2000*), *Gt(ROSA)26Sor*[tm14(CAG-tdTomato)Hze] referred to as *R26R*[TdT] (*Madisen et al., 2010*), and *Gt(ROSA)26Sor*[tm4(ACTB-tdTomato,-EGFP)Luo] referred to as *R26R*[mTmG] (*Muzumdar et al., 2007*). *Srf*[FLAG] and *Srf*[al] mice were generated by gene targeting. Homology arms of 2 kb and 6.4 kb were cloned into the pPGKneoF2L2DTA backbone. The longer arm was assembled in three fragments using HiFi assembly cloning (NEB) and included a 3 x FLAG tag introduced with a primer. Fragments were

amplified from 129S4 genomic DNA using Q5 polymerase (NEB) except for the middle segment of the long arm (i.e. the coding sequence of exon 1), which was amplified from a cDNA clone before or after introducing the αI helix mutations via site-directed mutagenesis. The targeting constructs were linearized and electroporated in AK7 (129S4 lineage) embryonic stem cells. Clones were selected with G418, screened by long-range PCR, and verified by Southern blot. Correctly targeted clones were injected into C57BL6/J E3.5 blastocysts, transferred to pseudopregnant F1 (C57BL6/J X 129S4) surrogates, and chimeras selected based on coat color. Founders were crossed to *MORE-Cre* mice to remove the *NeoR* cassette (*Tallquist and Soriano, 2000*). All mice were analyzed on a 129S4 co-isogenic background. Genotyping primers are available in *Supplementary file 2*.

## Conservation

Representative species from the various taxa were subject to BLASTP searches with default parameters using the amino acid sequences for mouse SRF, MKL2, and ELK1. Potential hits were then confirmed by reciprocal BLASTP back to mouse. If a species lacked an ELK1 homolog, the mouse ETS1 sequence was used to search for ETS-domain containing genes. If the given species lacked a hit for a particular search, the search was repeated for the entire taxon. Ctenophora were searched using amino acid sequences for mouse SRF and MEF2C, yeast MCM2, snapdragon Deficiens, and *Arabidopsis* Agamous. No homolog was identified, suggesting loss of MADS proteins in this lineage.

## RNA sequencing

FNPs (LNP+ MNP) and mandibles were carefully removed from E11.5 embryos in ice-cold PBS using fine forceps. A total of eight embryos across two litters representing four mutants and four controls were collected. Total RNA was immediately extracted using the RNeasy Plus Mini kit (Qiagen). RNA quality was assessed by Tapestation and all samples had RIN scores ≥ 9.8. Samples were sent to GeneWiz for PE150 sequencing. There were 14.6–30 m reads / sample and an average of 24.2 m reads / sample.

Reads were pseudo-aligned to the mouse transcriptome (mm10 partial selective alignment method, downloaded from refgenie) using salmon 1.5.0 and the flags –validateMappings –gcBias –numBootstraps 30. Pseudoalignments were processed with wasabi 1.0.1 and analyzed with sleuth 0.30.0–4 with the flag gene_mode = true. Analysis was performed using a full model that accounted for genotype, litter/batch, and tissue-of-origin (for combined tissue analysis only) versus a reduced model consisting only of litter/batch (and tissue-of-origin). Fold-changes and q-values were computed using the Wald test. Volcano plots were made with VolcaNoseR. Heat maps were generated using the Shinyapp HeatMappr. Gene set enrichment analysis for GO terms, ENCODE datasets, etc. were done with the web utility Enrichr (*Xie et al., 2021*). Enrichment for a custom list of targets was performed using GSEA software 4.10 and normalized read counts for the entire dataset. Analysis for the joint tissue model was run in phenotype mode ( > 7 samples per condition) and for the individual tissue samples in gene_set mode ( < 7 samples per condition) according to the software developer. The maximum number of genes per set was raised to 800 to accommodate the target lists. All other parameters were default.

## MEPM culture

Mouse embryonic palatal mesenchyme cells were generated as described (*Fantauzzo and Soriano, 2017*). Briefly, palatal shelves were dissected from E13.5 embryos in ice cold PBS using fine forceps. Yolk sac tissue was used for genotyping. Palates from individual embryos were held on ice until dissection was complete and palates were then dissociated using 0.125% Trypsin-EDTA at 37°C for 10 min with occasional trituration using a P1000 pipet. Trypsin was neutralized with an equal volume of growth media (DMEM High Glucose supplemented with Glutamine, Penicillin-Streptomycin, and 10% Fetal Calf III serum) and plated onto culture dishes coated in 0.1% gelatin. Cells were passaged as they approached confluency, every 2–3 days, and used for experiments at passage 2. MEPM cell lines were not screened for mycoplasma as they were used and extinguished by passage 2, but there was no evidence of mycoplasma by DAPI staining and other immortalized cell lines used in the lab tested negative for mycoplasma, so the risk of contamination was negligible.

## Immunofluorescence

MEPM cells were seeded on #1.5 coverslips coated with 0.1% gelatin. For starvation experiments, cells were starved overnight in 0.1% serum then stimulated 30' with 10% serum. Cells were fixed using 4% PFA in PBS for 10' at 37 °C. Embryos were dissected in ice cold PBS, fixed one hour in 4% PFA in PBS at 4 °C, rinsed in PBS, cryoprotected in 30% sucrose, and embedded in OCT. Sections were cut at 10 μm thickness using a Leica cryostat. Yolk sacs were fixed one hour in 4% PFA in PBS at 4 °C and stained whole. All samples were rinsed in PBS, blocked and permeabilized in blocking media (PBS, 0.3% TritonX-100, 1% BSA, 5% calf serum) one hour at RT, primary antibody was diluted in fresh blocking media and samples treated overnight at 4 °C, washed 3 x PBS at RT, incubated in Alexa Fluor Plus-conjugated secondary antibodies (Invitrogen) diluted 1:500 in fresh blocking media with 1 μg/ml DAPI for 1–2 hr at RT or overnight at 4 °C, and finally washed 3 x in PBS at RT. Samples were mounted in Prolong Diamond (Invitrogen) mounting media and imaged on a Zeiss AxioObserver inverted fluorescence microscope or a Zeiss 780 upright confocal microscope. Thresholding was performed and scalebars added in the FIJI implementation of ImageJ. All images for a given experiment were processed identically with the exception of *Figure 2A* where the mutant embryos were brightened compared to the control embryo to better illustrate phenotypes.

Quantitation of cell proliferation and cell death was performed by staining frozen sections with the indicated antibodies. Sections at the level of the heart were imaged using a 10 x objective on a Zeiss 780 confocal microscope, a 1024 × 1024 pixel count and 6 μm step size. Tiling was used with 10% overlap when necessary to image the entire section. Maximum intensity projections were made in the FIJI implementation of ImageJ and identical thresholds used for each embryo to calculate the DAPI-positive and cleaved Caspase 3-positive or phospho-Histone H3 (Ser 10)-positive area on sections at the level of the heart. Any of the embryo's posterior present in the section was ignored as this region was not present in all sections for all embryos.

Antibodies used were, rat anti-CD31 (BD Pharmingen, 553370) 1:50, rabbit anti-cleaved Caspase 3 (Cell Signaling 9665, 1:400), rabbit anti-phospho Histone H3 (Ser10) (Millipore 06–570, 1:500), rabbit anti-MKL1 (Proteintech, 21166–1-AP) 1:100, rabbit anti-SMA (Cell Signaling, 19245) 1:200. Phalloidin-Alexa Fluor 647 (Invitrogen) was included where indicated during secondary antibody staining at 1:400.

## RT-qPCR

Cells were seeded, passaged, starved, and stimulated as for immunofluorescence except in 12-well tissue culture plates. Embryo facial prominences were dissected in cold PBS and transferred to 1.5 ml microfuge tubes on ice. Following the indicated stimulation regimes when applicable, cells/tissue were lysed in 300 μl RLT buffer supplemented with BME, and RNA isolated using the RNeasy Plus Kit (Qiagen) according to the manufacturer's instructions. RNA concentration was quantified using a Nanodrop. One μg total RNA was used for reverse transcription. RNA was primed using a 2:1 molar ratio of random hexamer and polydT (Invitrogen) and reverse transcribed with Superscript IV (Invitrogen) according to the manufacture's protocol. Resulting cDNA was diluted 5 x with water and stored at –20 °C. One μl cDNA was used per qPCR reaction. qPCR was performed using Luna 2 x Master Mix (NEB) on an iQ5 thermocycler (Bio-Rad) in triplicate. Differences in gene expression were calculated by ΔΔCT using *Hprt* for normalization. Primer sequences are listed in *Supplementary file 3*.

## Nuclear translocation analysis

Cells were starved, treated, and stained as described above and imaged on an inverted Zeiss AxioObserver microscope. Z-stacks were maximum intensity projected in the FIJI implementation of ImageJ, background subtracted, and the DAPI channel used to create a nuclear mask. This mask was then used to measure the average nuclear intensity in the MRTF-A channel for each nucleus. The data presented are the pooled results from two cell lines of each genotype where each dot is an individual nucleus. At least 70 cells were analyzed per condition.

## In situ hybridization

E10.5 embryos were dissected in ice-cold PBS and fixed overnight in 4% FA in PBS at 4 °C, rinsed in PBS, dehydrated through a MeOH series and stored in 100% MeOH at –20 °C. Embryos were stained using standard techniques for the indicated transcripts using published, DIG-labeled probes,

and were developed in BM Purple (Roche | Sigma-Aldrich). For *Fgf8* the proteinase-K digestion was omitted in order to maintain integrity of the ectoderm. A detailed protocol is included as **Supplementary file 2** and probe sequences are available as SnapGene files at Dryad. https://doi.org/105061/dryadmgqnk9916.

## Histology

P0 hearts were fixed overnight in 4% FA in PBS at 4 °C, rinsed in PBS, dehydrated through an ethanol series, and embedded in paraffin. Five μm sections were cut using a Leica microtome. After drying, sections were stained with Harris modified hematoxylin (Fisher) and Eosin Y using a standard regressive staining protocol.

## Skeletal preparations

Skeletons were stained by standard techniques. Briefly, E18.5 embryos were skinned, eviscerated, fixed in ethanol, stained with.015% alcian blue and.005% alizarin red overnight at 37 ° C, cleared in 1% KOH, processed through a glycerol:KOH series, and photographed in 80% glycerol in PBS.

## Statistical methods

Specific statistical methods, significance values, and n are detailed in the figure legends. For RNA-Seq, statistics were computed using the built-in Wald Test function in the Sleuth analysis package. All other statistics were performed using GraphPad Prism 9.

## Acknowledgements

We thank the members of the Soriano lab, Robert Krauss, Sergei Sokol, and Harish Vasudevan for critical reading of the manuscript. We are appreciative of Kevin Kelley and the Mouse Transgenic Core for stable tissue culture facilities. We thank the Mt. Sinai Microscopy core facility for consistent advice and top-notch instrumentation. CJD was supported by F32 DE026678 from National Institutes of Health (NIH)/National Institute of Dental and Craniofacial Research (NIDCR). This work was supported by R01 DE022363 from NIH/NIDCR to PS.

## Additional information

### Funding

| Funder | Grant reference number | Author |
| --- | --- | --- |
| National Institute of Dental and Craniofacial Research | R01 DE022363 | Philippe Soriano |
| National Institute of Dental and Craniofacial Research | F32 DE026678 | Colin J Dinsmore |

The funders had no role in study design, data collection and interpretation, or the decision to submit the work for publication.

### Author contributions

Colin J Dinsmore, Conceptualization, Data curation, Formal analysis, Funding acquisition, Investigation, Methodology, Project administration, Resources, Validation, Visualization, Writing - original draft, Writing - review and editing; Philippe Soriano, Conceptualization, Funding acquisition, Investigation, Project administration, Supervision, Writing - review and editing

### Author ORCIDs

Colin J Dinsmore http://orcid.org/0000-0002-6404-1202
Philippe Soriano http://orcid.org/0000-0002-0427-926X

### Ethics

All animal experimentation was conducted according to protocols approved by the Institutional Animal Care and Use Committee of the Icahn School of Medicine at Mount Sinai under protocol

LA11-00243. Mice were kept in a dedicated animal vivarium with veterinarian support. They were housed on a 13hr-11hr light-dark cycle and had access to food and water ad libitum.

### Decision letter and Author response
Decision letter https://doi.org/10.7554/eLife.75106.sa1
Author response https://doi.org/10.7554/eLife.75106.sa2

## Additional files

### Supplementary files
• Supplementary file 1. RNA-Seq gene lists. A Microsoft Excel file containing gene expression analysis for the mandible, FNP, and joint mandible+ FNP datasets, FNP vs mandible dataset, gene lists used to generate *Figure 1G–H*, raw counts and TPM values for each sample, the variables used for each sample to classify it for analysis in Sleuth (genotype, tissue, and litter) and the Enrichr results used for *Figure 1F* and *Figure 1—figure supplement 2G-H*.

• Supplementary file 2. Knockout Phenotypes. Summary of the mouse knockout phenotypes for *Srf* and its cofactors, as well as a selection of relevant conditional knockouts discussed in the Introduction and Discussion sections.

• Supplementary file 3. Genotyping Primers. A list of genotyping primers and product sizes. All reactions were run for 35 cycles with an annealing temperature of 60 ° C.

• Supplementary file 4. qPCR Primers. Forward and reverse primer sequences used for qPCR experiments in *Figure 4*. All primers are listed 5' to 3'.

• Transparent reporting form

### Data availability
The NGS data is available on GEO. https://www.ncbi.nlm.nih.gov/geo/query/acc.cgi?acc=GSE186770.

The following dataset was generated:

| Author(s) | Year | Dataset title | Dataset URL | Database and Identifier |
|---|---|---|---|---|
| Dinsmore CJ, Soriano P | 2021 | Differential regulation of cranial and cardiac neural crest by Serum Response Factor | https://www.ncbi.nlm.nih.gov/geo/query/acc.cgi?acc=GSE186770 | NCBI Gene Expression Omnibus, GSE186770 |

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
