## [Editor Report]

This carefully executed study demonstrates new mechanisms by which Serum Response Factor (Srf) regulates transcription. The authors report the effects that loss of Srf function has on different neural crest lineages in the mouse. The results convincingly show that the main function of Srf within neural crest is in the cardiac neural crest lineage where it regulates cytoskeletal genes.

---

## [Decision Letter]

**Decision letter after peer review:**

Thank you for submitting your article "Differential regulation of cranial and cardiac neural crest by Serum Response Factor" for consideration by *eLife*. Your article has been reviewed by 3 peer reviewers, and the evaluation has been overseen by Marianne Bronner as the Senior and Reviewing Editor. The following individual involved in review of your submission has agreed to reveal their identity: Richard Treisman (Reviewer #2).

Essential revisions:

All three reviewers expressed great enthusiasm for this study and feel that it will make a valuable contribution to the literature. That said, they also include extensive suggestions for improvements and clarifications to the text and figures. While these suggestions are numerous, all of the proposed changes are editorial and do not require further experimentation. I refer you to the comments of the reviewers below for details on suggested modifications.

*Reviewer #1 (Recommendations for the authors):*

1) Page 4: it would be nice to show a supplementary a table with the summary of the phenotypes of the co-factor knock outs.

2) Figure 1B: Please label upper and lower panels with mutant and sibs. The difference in labeling (flox/+ and flox/flox is difficult to see).

3) Please provide an excel file with the genes that are associated with the different Enrichment terms.

4) Page 9: 'Underscoring their importance, we found these residues are conserved in Srf homologs from human to sponge, though they are intriguingly less well-conserved in clades lacking a readily identifiable Mrtf ortholog'.

Please add: '.., such as… (list the clades)'.

5) Page 11: In Srf^α^ mutants the authors detected an increase in b-Catenin levels. Please discuss this finding, as there are studies published regarding the interaction between Srf and Wnt signaling.

6) Page 13, line 285. Please add a sentence describing the interpretation of the findings in this paragraph.

7) Line 398: typo: but not does…

8) In the same paragraph it is unclear if starved is compared to serum, or mutant to sibling. Please clarify.

9) How do the authors explain that the gene expression changes are the same between the null and the point mutation mutant but the phenotypes are different? Are very subtle gene expression changes enough to cause the phenotypic differences?

*Reviewer #2 (Recommendations for the authors):*

I strongly support publication of this paper, with some modifications. Specific points to address:

Figure 1. This revisits the 2014 GD data for the Wnt1-cre SRF knockout. Its real value for this study would be to act as a control for analysis of the Wnt1-cre SRF-alpha1 mutant, but this is not done. This is important, because it may give insights into potential gene expression differences between the null and SRF-alpha1. The authors should do this experiment.

Figure 4. While serum stimulation is of interest, a more pertinent analysis, given the fact that there is no genetic interaction bewteen SRF-alpha1 and PDGFR, would be to look at PDGF stimulation, either in this system or MEPMs. What happens? Are there differences between Wnt1-cre SRF KO and SRF-alpha1?

*Reviewer #3 (Recommendations for the authors):*

1. The authors suggest in the discussion that TCF factors may compensate for loss of SRF-MRTF activity. Could the authors provide some in vivo staining to show the expression patterns of genes involved in these two complexes? Although bulk-RNAseq data shows that these two complexes are both involved in samples collected from the craniofacial region, it's hard to conclude whether these two complexes could compensate for each other without information about their in vivo expression patterns.

2. Defects of the neural tube are not visible in Figures 2A and 2C. Could the authors adjust the orientation of the samples for better visualization and clarification of the phenotype?

3. Since MRTF-A translocated to the nucleus in response to serum in control and mutant cell lines, the authors need to provide explanation as to why there are no differences observed in control groups with/without serum?

4. Bulk RNA-seq analysis using tissue from Srf^flox/flox^;Wnt1-Cre^Tg/+^ mice strongly suggests that the SRF-MRTF complex plays a critical role in craniofacial development. Therefore, the authors generated Srf αl mice, in which they mutated the αI helix of the SRF DNA-binding domain to disrupt SRF-MRTF complex formation while leaving SRF-TCF-DNA complex formation unaffected. However, in the neural crest conditional mutant (Srf^αI/flox^;Wnt1-Cre^Tg/+^) mice, the defects were detected not in the craniofacial tissues but in cardiac development. Could the authors provide more explanation for these results? Are there downstream genes targeted by both SRF-MRTF and SRF-TCF in the craniofacial tissues so that the SRF-TCF complex can compensate for the lost function of the SRF-MRTF complex?

5. Why did the authors study the function of SRF-MRTF in the mesoderm-derived lineage? Is this an attempt to show the efficiency of the Srf^αI/al^ model? Do Srf^flox/flox^;Wnt1-Cre^Tg/+^ mice have outflow tract defects?

---

## [Author Response]

Reviewer #1 (Recommendations for the authors):1) Page 4: it would be nice to show a supplementary a table with the summary of the phenotypes of the co-factor knock outs.

We thank the reviewer for this suggestion and now include these phenotypes as Supplementary Table 1. The other tables have been renumbered accordingly.

2) Figure 1B: Please label upper and lower panels with mutant and sibs. The difference in labeling (flox/+ and flox/flox is difficult to see).

We have clarified the labeling in this panel as well as panel 1E.

3) Please provide an excel file with the genes that are associated with the different Enrichment terms.

These data are now included as tabs in the Supplementary File. We regret our earlier omission.

4) Page 9: 'Underscoring their importance, we found these residues are conserved in Srf homologs from human to sponge, though they are intriguingly less well-conserved in clades lacking a readily identifiable Mrtf ortholog'.Please add: '.., such as… (list the clades)'.

We now highlight the pertinent clades in the manuscript and thank the reviewer for the suggestion.

5) Page 11: In Srf^α^ mutants the authors detected an increase in b-Catenin levels. Please discuss this finding, as there are studies published regarding the interaction between Srf and Wnt signaling.

We also found this quite interesting and worthy of future study but were unsure how broadly to speculate on its implications within the manuscript. One could imagine the change in beta-catenin levels to be caused through direct transcriptional regulation, through altered cytoskeletal behavior (speculatively, by increased recruitment and/or stabilization at junctions in an attempt to compensate for reduced F-actin levels or through altered cadherin levels), or by changes in beta-catenin protein’s stability, most obviously through a change in canonical Wnt activity. Increased canonical Wnt signaling should lead to beta-catenin stabilization or alternately, increased beta-catenin stabilization might ectopically activate or increase sensitivity to Wnt. We note that the enhanced beta-catenin expression remained enriched at cell junctions and did not build up within the nucleus. Further, we did not detect a strong Wnt signature in our RNA-seq studies. However, we realize nuclear beta-catenin levels can be difficult to detect (*Apc* mutants can show stabilized junctional, not nuclear, beta-catenin, e.g. Buchert et al., 2010) and a Wnt phenotype may be specific to the *Srf^αI^* allele or the early embryo and not neural crest. However, the existence, strength, and mechanism of this effect in the context of our *Srf* mutant remains unclear. Several reports have suggested a link between *Srf* and the Wnt pathway including the suppression of SRF-mediated IEG expression by Wnt, Wnt regulation of *Mrtfa* expression, SRF-mediated regulation of Wnt pathway targets, and phosphorylation of SRF by GSK3 promoting SRF-MRTF activity. Most intriguingly, one study found an effect on Wnt through SRF-mediated expression of a microRNA that regulates EMT and E-Cadherin, indirectly affecting beta-catenin localization. However, after considering these numerous potential explanations, we feel this result might be more distracting than helpful in the present manuscript. We hope to include this result in a future manuscript where the connections between SRF, adherens junctions, beta-catenin, and potentially the Wnt pathway can be explored – and tested – in greater detail. Therefore, we have opted to remove those two panels from the present manuscript.

6) Page 13, line 285. Please add a sentence describing the interpretation of the findings in this paragraph.

We added a clause explaining that we expected MRTF-A’s localization to be unaffected by our mutations in *Srf*. The upstream actin dynamics and signaling pathways that regulate MRTFs should remain intact in our point mutant background. There was a formal possibility that mutating *Srf* could cause an indirect change in the pathways that control MRTF localization. If this were true, we might interpret transcriptional changes as being due to the failure of SRF^αI^ to interact with MRTF, when in fact it may be due to the unexpected failure of MRTF to translocate to the nucleus at all. We agree that in the original manuscript, ending the paragraph with a rather bluntly (un)explained control experiment was confusing.

7) Line 398: typo: but not does…

Fixed. We thank the reviewer for their careful reading.

8) In the same paragraph it is unclear if starved is compared to serum, or mutant to sibling. Please clarify.

We were unsure which experiment this comment was referring to, but if it is in reference to the residual 5-10% complex formation discussed later in the same paragraph, that experiment was comparing wild type to mutant protein in an in vitro complex formation (gel shift) assay. We added further description of the experiment to clarify this matter.

9) How do the authors explain that the gene expression changes are the same between the null and the point mutation mutant but the phenotypes are different? Are very subtle gene expression changes enough to cause the phenotypic differences?

We thank the reviewer for highlighting this confusing point. We found strong phenotypic and gene expression changes in cultured MEPM cells derived from the *Srf^αI^* conditional mutants (Figure 4). *Srf^flox/flox^* conditional (*Wnt1-Cre*) mutants do not survive long enough to culture similar MEPM cells. The gene expression experiments in Figure 1 were from E11.5 craniofacial tissue. To conduct a comparison of these two genetic models using an identical tissue source and assay, we performed RT-qPCR from E11.5 *Srf^αI/flox^* and *Srf^flox/flox^ Wnt1-Cre* conditional mutants and littermate controls. We find that the cytoskeletal genes we assayed were downregulated in both models, but more strongly in the conditional knockouts than in the conditional point mutants. This likely explains the different phenotypes: there is one threshold for failure in the craniofacial crest (and possibly also the gastrula-stage embryo) and another for failure in the cardiac crest (which forms smooth muscle) and the *Mesp1* mesodermal lineage. However, whether this expression is due to residual MRTF activity or transcription of the same targets supported by other cofactors remains to be determined. We have now added these important data as figure 4E-F and commented where relevant, although these data ultimately do not substantially change the possible mechanistic interpretations covered in the Discussion.

Reviewer #2 (Recommendations for the authors):I strongly support publication of this paper, with some modifications. Specific points to address:Figure 1. This revisits the 2014 GD data for the Wnt1-cre SRF knockout. Its real value for this study would be to act as a control for analysis of the Wnt1-cre SRF-alpha1 mutant, but this is not done. This is important, because it may give insights into potential gene expression differences between the null and SRF-alpha1. The authors should do this experiment.

This is an excellent suggestion. The *Srf^lox/lox^; Wnt1-Cre* mice do not survive long enough on our genetic background to generate MEPMs and compare directly with the cells in Figure 4. We were also concerned that repeating the RNA-Seq analysis on the conditional point mutants would take an excessive amount of time. However, we agree that a side-by-side assessment of the two models in NC at the level of gene expression, rather than phenotype, was warranted and therefore performed RT-qPCR on several of the most differentially regulated genes from our RNA seq on freshly isolated E11.5 craniofacial prominences from both conditional point mutant and conditional knockout embryos (Vasudevan and Soriano, 2014). We find a reduction in gene expression in both models, but the effect is roughly 2x more profound in the conditional knockouts. We think this is consistent with the observed phenotype. The residual gene expression must be sufficient in craniofacial NC (and perhaps gastrula) but not in *Mesp1* or cardiac NC lineages. The qPCR data are now included as Figure 4E-F and the Discussion has been updated.

Figure 4. While serum stimulation is of interest, a more pertinent analysis, given the fact that there is no genetic interaction bewteen SRF-alpha1 and PDGFR, would be to look at PDGF stimulation, either in this system or MEPMs. What happens? Are there differences between Wnt1-cre SRF KO and SRF-alpha1?

This is an interesting suggestion, but a difficult experiment to do. The *Srf* NC conditional mutants do not survive long enough on our genetic background to generate MEPMs. Some experiments were performed in our previous study using cultured *Srf* conditional knockout cells derived from an earlier timepoint, but these were only immunofluorescent experiments on a small number passage 0 cells, as these cells did not take well to culture. The experiment could best be done by crossing the relevant alleles into an immortalized background such as *Ink4a* or generating *Srf* null and point mutant cell lines in an already immortalized but otherwise wild type cell line via CRISPR. However, generating these materials would take a longer amount of time than can be expected for the manuscript revision.*Reviewer #3 (Recommendations for the authors):*

1. The authors suggest in the discussion that TCF factors may compensate for loss of SRF-MRTF activity. Could the authors provide some in vivo staining to show the expression patterns of genes involved in these two complexes? Although bulk-RNAseq data shows that these two complexes are both involved in samples collected from the craniofacial region, it's hard to conclude whether these two complexes could compensate for each other without information about their in vivo expression patterns.

We thank the reviewer for raising this point. Some of these data were presented in a previous publication (Vasudevan and Soriano, 2014). We also now include previously published RNA-Seq data on sorted NC cells, whole palate, and passage 2 MEPM cells showing co-expression of *Srf* and its cofactors in these cells and tissues in Figure 1 Supplement 2E, as well as our own bulk RNA seq data for the same genes, showing a similar pattern. We attempted to perform immunofluorescent staining for cofactors in frozen sections but did not obtain convincing staining, possibly due to the lower relative expression levels of ELK1 and MRTFA, which have the most well-validated antibodies.

2. Defects of the neural tube are not visible in Figures 2A and 2C. Could the authors adjust the orientation of the samples for better visualization and clarification of the phenotype?

We agree the images were rather dim and small in the initial version of the figure. Therefore, we have first readjusted the brightness of the mutant embryos in figure 2A. They were initially imaged and thresholded identically to the control, but as they are smaller with fewer cells, this resulted in a dimmer image. We readjusted them (and added a note to the Methods section) and added arrowheads pointing out the phenotypes. We think the open neural tube/exencephaly and wavy neural tube are now much easier to see. Second, we added a new set of panels where two of the E9.5 embryos were reimaged at a higher magnification using a confocal microscope. The pharyngeal arch defect and open midbrain are now much clearer. We thank the reviewer for the suggestion.

3. Since MRTF-A translocated to the nucleus in response to serum in control and mutant cell lines, the authors need to provide explanation as to why there are no differences observed in control groups with/without serum?

See response to reviewer 1, point 6. Briefly, MRTF translocation should be unaffected, but once in the nucleus it should have reduced or absent formation of a productive SRF-MRTF-DNA ternary complex with our SRF point mutant compared to wild type SRF. The experiment was a control that nothing upstream was unexpectedly affected. We added clarification in the manuscript on this point.

4. Bulk RNA-seq analysis using tissue from Srf^flox/flox^;Wnt1-Cre^Tg/+^ mice strongly suggests that the SRF-MRTF complex plays a critical role in craniofacial development. Therefore, the authors generated Srf αl mice, in which they mutated the αI helix of the SRF DNA-binding domain to disrupt SRF-MRTF complex formation while leaving SRF-TCF-DNA complex formation unaffected. However, in the neural crest conditional mutant (Srf^αI/flox^;Wnt1-Cre^Tg/+^) mice, the defects were detected not in the craniofacial tissues but in cardiac development. Could the authors provide more explanation for these results? Are there downstream genes targeted by both SRF-MRTF and SRF-TCF in the craniofacial tissues so that the SRF-TCF complex can compensate for the lost function of the SRF-MRTF complex?

We performed RT-qPCR testing of *Srf^αI/flox^;Wnt1-Cre^Tg/+^* and *Srf^flox/flox^;Wnt1-Cre^Tg/+^* to get a better comparison of gene expression differences in NC among the two alleles and find downregulation of actin cytoskeletal genes in both models, but to a lesser degree in the point mutant compared to the conditional knockout. Whether this is due to TCF compensation (it has been suggested that TCF and MRTF coregulate some targets), residual MRTF function, and/or TCF/MRTF-independent SRF function is unknown at this time, but we discuss these possibilities within the Discussion. We have reorganized the Discussion section to incorporate ideas raised by reviewer 2 and we hope this makes these points more clearly. Reviewer 2 suggested several excellent experiments to test these ideas including generation of an *Srf^V194E^* allele that should disrupt both TCF and MRTF interactions. Another good test would be conditional ablation of *Mrtfa/b* in the NC. We are excited to pursue these possibilities going forward but agree with reviewer 2 that these experiments are beyond the scope of this manuscript. Finally, to address the reviewer’s question concerning craniofacial vs cardiac phenotypes, it seems likely given the strong effect in the *Mesp1* lineage and SRF’s known role in muscle that the *Srf^αI^* allele has a particularly strong effect in the cardiac crest because these cells mostly differentiate into smooth muscle, whereas craniofacial NC cells become other cell types such as bone, cartilage, neurons, glia, melanocytes, and dermal mesenchyme. Therefore, a threshold effect is likely at work where cells requiring high levels of actin-related gene expression (i.e. muscle) are more affected by *Srf^αI^* than those that require more modest expression levels. Further testing of this idea with new genetic models will be key.

5. Why did the authors study the function of SRF-MRTF in the mesoderm-derived lineage? Is this an attempt to show the efficiency of the Srf^αI/al^ model? Do Srf^flox/flox^;Wnt1-Cre^Tg/+^ mice have outflow tract defects?

The reviewer is correct. Because vascular and cardiac mesoderm are known to rely on SRF-Myocardin activity, we wanted to test the efficiency of the *Srf^αI^* allele compared to the *Srf^-^* allele (and published mutants such as *Myocd^-/-^*) in this context. Further, we wondered if the wavy neural tube was (at least partially) non-autonomous and due to mesoderm defects. We did see this phenotype in *Srf^αI/flox^; Mesp1^Cre/+^* embryos at E9.5, supporting this interpretation. We failed to mention this observation in the initial manuscript and have now added this rationale and finding to the Results section.

*Srf^flox/flox^; Wnt1-Cre* mice do have cardiac outflow tract defects, as reported by Newbern et al. We mention this finding within the Discussion.